

# Exploring the role of individual learning in animal tool-use

Elisa Bandini and Claudio Tennie

Department of Prehistory and Quaternary Ecology, Eberhard-Karls-Universität Tübingen, Tübingen, Germany

## ABSTRACT

The notion that tool-use is unique to humans has long been refuted by the growing number of observations of animals using tools across various contexts. Yet, the mechanisms behind the emergence and sustenance of these tool-use repertoires are still heavily debated. We argue that the current animal behaviour literature is biased towards a social learning approach, in which animal, and in particular primate, tool-use repertoires are thought to require social learning mechanisms (copying variants of social learning are most often invoked). However, concrete evidence for a widespread dependency on social learning is still lacking. On the other hand, a growing body of observational and experimental data demonstrates that various animal species are capable of acquiring the forms of their tool-use behaviours via individual learning, with (non-copying) social learning regulating the frequencies of the behavioural forms within (and, indirectly, between) groups. As a first outline of the extent of the role of individual learning in animal tool-use, a literature review of reports of the spontaneous acquisition of animal tool-use behaviours was carried out across observational and experimental studies. The results of this review suggest that perhaps due to the pervasive focus on social learning in the literature, accounts of the individual learning of tool-use forms by naïve animals may have been largely overlooked, and their importance under-examined.

## INTRODUCTION

*'The natures of animals are untutored'*
    -Hippocrates (Epidemics vI, 32)

A large number of non-human animal species (henceforth: animals) use tools (tool-use is defined as: 'The external employment of unattached or manipulable attached environmental object to alter more efficiently the form, position or condition of another object, another organism, or the user itself, when the user holds and directly manipulates the tool during or prior to use and is responsible for the proper and effective orientation of the tool'; *Shumaker et al., 2011*; see also reviews within and *Beck, 1980*; *Bentley-Condit & Smith, 2010*; *Finn, Tregenza & Norman, 2009*; *Kenward et al., 2011*; *Krützen et al., 2005*; *Ottoni & Mannu, 2001*; *Robbins et al., 2016*; *Whiten et al., 1999*). Within tool-using animals, a smaller subset of species use tools in flexible ways, creating repertoires of tool-use behaviours that can vary both within and between populations

Corresponding author
Elisa Bandini,
elisa-bandini@hotmail.it

**Table 1** Definitions of relevant social learning mechanisms and factors used throughout this review.

| Term | Definition |
|---|---|
| Behavioural form | The specific action component(s) and organisation of a behaviour. Can be organised in a linear and/or hierarchical relationship. |
| Artefact form | The specific physical component(s) and configuration(s) as outcomes of behaviour. Can be organised in a linear and/or hierarchical relationship. |
| Copying social learning | Mechanisms that transmit the actual form of a behaviour and/or artefact (must be in a causal relationship between original and copy). Transmits 'know-how'. Includes social learning mechanisms such as end-state emulation and imitation. |
| End-state emulation | An individual learns about the environmental affordances and the products of a behaviour, and causally reproduces a similar end-state but in doing so applies their own strategies to produce the end-state (these strategies may or may not match the originally used strategies; *Tomasello, 1996*). |
| Imitation | An individual copies the form of a behaviour (compare also *Galef, 1998*). |
| Non-copying social learning | Mechanisms that do not transmit the form of a behaviour or artefact. Instead, these mechanisms often regulate the frequencies of forms (e.g. by increasing a subject's motivation to interact with certain objects and/or locations). Typical mechanisms here include local ('know-where') and stimulus enhancement ('know-what'). |
| Local enhancement | The salience of a location is enhanced by other individuals being at or interacting with or near that location (compare *Hoppitt & Laland, 2013*; *Tennie, Hopper & Van Schaik, 2020*) |
| Stimulus enhancement | The salience of an object is enhanced by other individuals interacting with this type of object (*Hoppitt & Laland, 2013*) |

(*Hunt, Gray & Taylor, 2013*). Yet it is still unclear how these animals acquire and sustain their tool-use repertoires.

Early studies on the mechanisms behind animal tool-use suggested that these behaviours were acquired primarily via individual learning (*Köhler, 1925* and see also *Bandini et al., 2020b*). Since then, almost all animal species have demonstrated an ability to use information via non-copying types of social learning, (i.e. social learning variants that do not transmit the actual form of a behaviour or artefact, such as local and/or stimulus enhancement etc.; *Akins, Klein & Zentall, 2002*; *Custance et al., 2001*; *Kendal et al., 2015*; *Kis, Huber & Wilkinson, 2015*; *Miller, Rayburn-Reeves & Zentall, 2009*; *Reader & Biro, 2010*; *Stoinski & Beck, 2001*; see Table 1 for definitions of the terms used throughout this review). These new findings stimulated a large research effort into the extent of social learning abilities across species, with a particular focus on non-human primates. Some researchers further hypothesised that because animals are influenced by non-copying variants of social learning, they could also copy behavioural and/or artefact forms (i.e. the action and/or physical components of a behaviour; see Table 1). This hypothesis has become most prevalent for chimpanzees and other primates, for which some authors have further argued that copying variants of social learning (i.e. social learning mechanisms that transmit the actual form of a behaviour or artefact, such as imitation; see Table 1) are necessary for the acquisition of some of their material culture (*Whiten et al., 1999*; *Lamal, 2002*; *Matsuzawa et al., 2008*; *Whiten & Van de Waal, 2017*). The emphasis on copying social learning in primate behaviour may be due, in part, to the close phylogenetic ties between humans and other primates and the fact that chimpanzees, in particular, have one of the most extensive tool-use repertoires (apart from humans) in the animal kingdom (*Whiten et al., 1999*). These ties between humans and other primates may have led some to also hypothesise that the mechanisms underlying non-human primate material culture

may be similar to that of modern humans, and therefore that they also rely to some degree on copying social learning. However, this 'copying hypothesis' has been, and still is, heavily debated (*Galef, 1992*; *Kendal, 2008*; *Laland & Galef, 2009*; *Tennie, Call & Tomasello, 2009*; *Tennie, Hopper & Van Schaik, 2020*; *Bandini & Tennie, 2017*; *2019*). The ability to copy behavioural and/or artefact forms has been suggested to be one of the key requirements for modern human culture (i.e. cumulative culture; *Boyd & Richerson (1996)*, especially of forms; *Reindl et al., 2016*; *Tennie, Hopper & Van Schaik, 2020*). Therefore, examining the specific mechanisms underlying animal culture, and whether any non-human species can copy behavioural and/or artefact forms will provide insight into whether copying social learning is a unique trait in modern humans, or whether it is shared with other animals.

This debate has driven researchers to investigate experimentally the ability of primates, alongside other animal species, to transmit and acquire form information via copying social learning mechanisms. So far, concrete evidence for the view that primates (or any other animals) strictly require copying to acquire their tool-use behavioural forms is still lacking. Indeed, past experimental approaches that examined the ability to copy behaviours in unenculturated primates were unable to find spontaneous form copying abilities in their subjects. Enculturated animals are those who have been exposed to extensive human contact and/or training—typically from a young age (e.g. language training; *Gardner, Gardner & Van Cantfort, 1989*). Enculturated animals are often hand-reared by humans and kept in a household, in some cases even being treated like human children (e.g. Kanzi the bonobo is an example of an enculturated animal; *Toth et al., 1993*). Whilst some may argue that captive animals in zoological institutions might have a higher potential level of enculturation over their wild counterparts as they are exposed to humans regularly, if (social) animals are kept in conspecific groups and have not received intensive human training or contact (and are mother-reared), they can be considered *unenculturated* (see *Henrich & Tennie (2017)* for further discussion). Thus, the current state of knowledge suggests that unenculturated primates, whilst attending to information via non-copying social learning, do not spontaneously copy behavioural forms (*Clay & Tennie, 2017*; *Tennie, Call & Tomasello, 2012*; *Pope et al., 2017*). An alternative view consistent with the currently available data suggests that animal tool-use forms are developed and sustained via an interplay of individual learning and non-copying social learning.

One approach in the literature, the Zone of Latent Solutions (ZLS) hypothesis (*Tennie, Call & Tomasello, 2009*), advocates for this explanation of animal tool-use behavioural repertoires. The ZLS proposes that many animal (and sometimes human) tool-use behavioural forms are within the 'Zone of Latent Solutions' of the species, and are acquired through individual learning. Regulated by non-copying social learning mechanisms. The frequency of latent solutions are however often regulated by non-copying social learning mechanisms which create and maintain the regional differences in repertoires the regional differences in repertoires observed in some wild animal populations (e.g. chimpanzees; *Whiten et al., 2001*, *1999*; orangutans; *Fox et al., 2004* and New Caledonian crows; *Hunt & Gray, 2003*). Therefore, according to the ZLS approach, observed tool-use repertoires are maintained via 'socially mediated reinnovations'

(SMR; *Bandini & Tennie, 2017*), in which each individual must reinnovate the behavioural forms from their ZLS (via individual learning), and are stimulated to do so via non-copying social learning (*Tennie, Call & Tomasello, 2009*).

Evidence for this approach stems from the growing number of reports of the spontaneous development of tool-use behaviours by naïve animals in the literature (e.g. Table 2). However, perhaps due to the general focus on social learning in the field, the relevance of these studies, and their implications for animal cognition, have been somewhat neglected in the current literature. Furthermore, to the best of our knowledge, a comprehensive review of reports of individual learning across animal species is lacking in the literature. Therefore, the aim of this review was to fill this gap by creating a database of studies on individual learning of tool-use behavioural forms across species (see Table 2). The database was created by collecting experimental and observational studies in which tool-use behavioural forms emerged spontaneously (i.e. the behaviour was first observed in naïve individuals who were most likely not exposed to demonstrations of the target behavioural form beforehand) in wild and captive animals. Most of the studies included in the database (Table 2) describe behavioural forms first observed in captive individuals, where the subject's previous knowledge and experience was known and could be controlled for. However, the database also includes observational accounts of the emergence of behaviours in the wild (where the authors had reason to believe that the innovator had not seen the behaviour beforehand, although note that these claims should be interpreted with caution as knowing the full behavioural repertoire of wild animals is difficult). This review therefore highlights the extent of individual learning in the tool-use repertoires of a wide range of species, suggesting that individual learning, supported by non-copying social learning plays an important (frequency-regulating) role in the acquisition and maintenance of animal tool-use repertoires.

## SURVEY METHODOLOGY

A literature search was carried out by EB between 2016 and 2019 using the terms: animal '*innovations*', '*inventions*', '*novel behaviour*', '*spontaneous*', '*individual learning*', '*problem-solving*' '*trial and error*' and '*tool-use*', following similar terms used in previous literature reviews on animal tool-use and innovation (*Reader & Laland, 2003*). Although the author was already aware of some of the individual learning studies before starting the review, these were only added into the database if they were found following the systematic approach described below. This procedure was carried out to ensure that the process was as unbiased as possible. The online search engines *Google Scholar* and *Web of Science* were used to search for relevant literature across journals. Once the literature search using these terms was exhausted, references from within the already accessed papers were examined to find ones that cited other relevant studies on the spontaneous expression of tool-use behavioural forms by naïve animals. Over 200 research papers were then read thoroughly by EB to ascertain, as best as possible, whether they (i) described the *spontaneous* (i.e. without copying required for the behavioural form) emergence of a behavioural form and (ii) that the individual who showed the behavioural form was naïve to the target behavioural form beforehand, and had not received any training or

exposure to the behaviour before testing or before the first observation (as described by the authors of the study). Studies were excluded if it could be inferred that the innovator had pre-existing knowledge of the behaviour that was reinnovated, or if it was clear that the innovator was enculturated (*Henrich & Tennie, 2017*). Additionally, we also excluded studies in which there was strong evidence that subjects had been deprived or living in deprived conditions when the observations were made, as although more data is required, it may be that deprivation influences the development of species-typical behaviours (*Neadle, Bandini & Tennie, 2020*). The studies that were included in the final database (Table 2) consist of reports of the first time a tool-use behavioural form was observed in a given population of wild animals; studies with captive subjects in which the development of a target tool behavioural form was the aim of the study; studies in which the main research aim was not to encourage individual learning, but reinnovations nonetheless occurred, and captive observational reports describing the emergence of behavioural forms.

Table 2 is divided by species, and includes the name of the species (column one), the reference (column two), the testing methodology (wild/captive and observational/experimental) applied in the study (column three), a brief summary of the main findings (column four), and interpretation of the findings by the authors (column five; quotes were taken directly from the papers to avoid any subjective interpretations by the author). No control for research effort or for the frequency of the species in the wild was carried out for this review, as the aim was to provide an overview of the individual learning abilities of various tool-using species. Therefore, it is likely that the resulting larger number of spontaneous tool-use reports found in primates and birds as opposed to other species is partially (if not primarily) a product of the research bias towards these species in the literature (*Sayers et al., 2008*).

## RESULTS

### Quantitative results

A total of 105 publications that fit the requirements described above were found by EB (see Table 1). Of these publications, 73.3% were on primates, including, in alphabetical order: baboons (*Papio papio; cynocephalus anubis*); bonobos (*Pan paniscus*); capuchins (*Cebus paella; capucinus; nigritus; libidinosus*); chimpanzees (*Pan troglodytes; schweinfurthii*); orangutans (*Pongo pygmaeus abelii*); gibbons (*Hylobatidae*); golden lion tamarins (*Leontopithecus rosalia rosalia*); gorillas (*Gorilla beringei beringei; gorilla gorilla)* and macaques (*Macaca tonkeana; fascicularis fascicularis; nemestrina; silenus; mulatta; fuscata*). 20.0% were publications on birds, including in alphabetical order: canaries (*Serinus canaria*); Goffin cockatoos (*Tanimbar corella*); Hawaiian crows (*Corvus hawaiiensis*); hyacinth macaws (*Anodorhynchus hyacinthinus*); kea parrots (*Nestor notabilis*); New Caledonian crows (*Corvus moneduloides*); Northern jays (*Cyanocitta cristata*); pigeons (*Columba livia domestica*); rooks (*Corvus frugilegus*) and woodpecker finches (*Cactospiza pallida*). 6.7% were on other animals, including in alphabetical order:

**Table 2 Studies on the individual learning abilities of tool-using animals.**

| Species | Reference | Testing methodology | Findings | Authors' interpretation |
|---|---|---|---|---|
| **Great Apes** | | | | |
| Bonobos (*Pan paniscus*) | *Boose, White & Meinelt (2013)* | Experimental study in captivity | The first individual to both attempt and successfully fish was a naïve juvenile female who investigated the mound, manufactured a tool, attempted, and successfully fished on the first baited trial presented to the bonobos. All the other members of the group eventually showed the same behaviour | 'Following *Lonsdorf et al. (2009)*, "investigation" of the mound was defined as using visual and olfactory senses to examine the contents of the bait holes in order to identify when subjects were first clearly aware that the termite mound contained bait. This definition appeared appropriate for most individuals, except for one. Maiko, a low-ranking male, did not investigate the mound under this definition, but was in the vicinity of fishing individuals on several occasions prior to his first attempt. In this single case, it appears that Maiko used information from observing the behaviours of other group members to identify that the mound contained bait' (p.923) |
| Bonobos (*Pan paniscus*) | *Roffman et al. (2015)* | Experimental study in captivity | All the bonobos in the group used modified branches and unmodified antlers or stones to dig under rocks and in the ground or to break bones to retrieve the food | 'Interestingly, the digging techniques used by PB and her son NY, who shared much time together, were very similar, possibly indicating social learning by NY. Likewise, some individuals in the WZ group (LB, BR, EJ) used tools occasionally, whereas others (MA, LA, LO, BZ, BO) did so rarely or not at all' (p.88) |
| Chimpanzees (*Pan troglodytes schweinfurthii*) | *Hobaiter et al. (2014)* | Observational study in the wild | Two chimpanzees independently demonstrated the first of the two novel behaviours (moss-sponging) and five individuals demonstrated the second novel behaviour (moss-sponge re-use). | 'Here, we tested both a static and a dynamic network model and found strong evidence that diffusion patterns of moss-sponging, but not leaf-sponge re-use, were significantly better explained by social than individual learning. The most conservative estimate of social transmission accounted for 85% of observed events, with an estimated 15-fold increase in learning rate for each time a novice observed an informed individual moss-sponging. We conclude that group-specific behavioural variants in wild chimpanzees can be socially learned, adding to the evidence that this prerequisite for culture originated in a common ancestor of great apes and humans, long before the advent of modern humans' (p.1) |
| Chimpanzees (*Pan troglodytes schweinfurthii*) | *Morgan & Abwe (2006)* | Observational study in the wild | First observation (albeit indirect) of population-wide nut cracking in Cameroon | 'This observation challenges the existing model of the cultural diffusion of nut-cracking behaviour by implying that it has been invented on multiple occasions' (p.1) |
| Chimpanzees (*Pan troglodytes schweinfurthii*) | *Yamamoto et al. (2008)* | Observational study in the wild | Spontaneous ant-fishing by one individual over an extended time period (two observations over 2 years). So far this behaviour has only been observed in this one individual. | 'The motivation to use tools may not only encourage young chimpanzees to socially learn transmitted tool use behaviours but may also lead them to innovate new tool use behaviours through individual exploration and trial and error leaning' (p.701) |
| Chimpanzees (*Pan troglodytes*) | *Alp (1997)* | Observational study in the wild | Eight chimpanzees were observed using sticks as foot and body protection against thorns while foraging | 'This form of tool-using is culturally unique to the Tenkere chimpanzees, as at other sites where these apes have been observed eating parts of kapok trees, there are no published records of this tool technology. In three of the stepping-stick tool use incidents, the chimpanzee used the tool(s), held between their greater and lesser toes, in locomotion. This form of tool use is the first recorded case of habitually used tools that can be justifiably categorised as being "worn" by any known wild population' (p.1) |

| Species | Reference | Testing methodology | Findings | Authors' interpretation |
|---|---|---|---|---|
| Chimpanzees (*Pan troglodytes*) | *Bandini & Tennie (2017)* | Experimental study in captivity | Two chimpanzees (one in each group) spontaneously reinnovated the same technique for scooping | 'Our results demonstrate that the wild form of scooping behaviour re-appeared independently in two naive chimpanzees (it was reinnovated twice). Thus, unlike human cumulative cultural behaviour, the observed patterns of scooping behaviour in the wild can be explained via Socially Mediated Serial Reinnovations (SMSR)' (p.13) |
| Chimpanzees (*Pan troglodytes*) | *Bandini & Tennie (2019)* | Experimental study in captivity | Three individuals spontaneously used tools to show the same pounding behavioural form | 'At least three individuals spontaneously reinnovated the stick pounding behaviour examined in this study (one individual in group one, two and four). In all three groups, the naïve chimpanzees used sticks and a pounding action to retrieve the bait at the bottom of the testing apparatus. These findings surpass the double-case ZLS standard and therefore suggest that stick pounding is a behaviour that can reinnovated via individual learning' (p.16) |
| Chimpanzees (*Pan troglodytes*) | *Bernstein-Kurtycz et al. (2020)* | Experimental study in captivity | Six chimpanzees used two different tools (one ridgid and one flexible) to retrieve a reward from an artifical termite mound | 'Overall, chimpanzees did use both tool types in order, thus demonstrating that this tool-set form is not a CDT. Yet, few attempts were successful, and the majority were not made using a tool set. This suggests that the behaviour did not stabilise, which may have been due to unintentional difficulty created by the opacity of the task. Overall, our study showed that the form of tool sets can be within the ZLS of chimpanzees, but future studies need to determine what stabilises the behavioural patterns in the wild' (p.1) |
| Chimpanzees (*Pan troglodytes*) | *Birch (1945)* | Experimental study in captivity | Two individuals used sticks to rake in the food in the first testing condition. Over the course of 3 days of exposure to the sticks, every one of the subjects succeeded in solving the problem within a period of 20 s | 'That insightful problem solution represents the integration into new patterns of activity of previously existent part-processes developed in the course of the animals' earlier activities' (p.382) |
| Chimpanzees (*Pan troglodytes*) | *Davis et al. (2016)* | Experimental study in captivity | Two naïve chimpanzees in the control group spontaneously discovered the more efficient method for removing the token from the puzzle box | 'When their foraging method became substantially less efficient, nine chimpanzees with socially-acquired information (four of whom witnessed additional human demonstrations) relinquished their old behaviour in favour of the more efficient one. Only a single chimpanzee in control groups, who had not witnessed a knowledgeable model, discovered this. Individuals who switched were later able to combine components of their two learned techniques to produce a more efficient solution than their extensively used, original foraging method. These results suggest that, although chimpanzees show a considerable degree of conservatism, they also have an ability to combine independent behaviours to produce efficient compound action sequences; one of the foundational abilities (or candidate mechanisms) for human cumulative culture' (p.2) |
| Chimpanzees (*Pan troglodytes*) | *Devos, Gatti & Levrero (2002)* | Observational study in the wild | First observation of spontaneous algae scooping with a tool by one chimpanzee | No interpretation offered by the authors |

(Continued)

| Species | Reference | Testing methodology | Findings | Authors' interpretation |
|---|---|---|---|---|
| Chimpanzees (*Pan troglodytes*) | Harrison & Whiten (2018) | Experimental study in captivity | The chimpanzees spontaneously used tools to access the juice in the first condition, however did not switch to more efficient techniques when access to the juice was restricted, even when sticks were placed inside the apparatus (providing a 'scaffolded condition') the subjects still did not adopt the more efficient technique | 'The limited exposure to scaffolding provided to four chimpanzees in the third and final phase of this study did not lead to the acquisition of novel techniques by any individual' (p.18) |
| Chimpanzees (*Pan troglodytes*) | Hirata, Morimura & Houki (2009) | Experimental study in captivity | During one of the pre-training tests, and after some exposure to the materials, the demonstrator (before being trained in the behaviour), spontaneously cracked open the nuts provided with the stones. In the pre-tests, one individual from the rest of the group also showed pre-cursors to nut-cracking by placing a nut on an anvil and hitting it with one of the stones. The attempts were never successful, however. The whole group eventually acquired the full behaviour, but this was after demonstrations had been carried out. | 'Another implication of this study is the effect of observing a model. It is difficult to prove the effect of observing a model in our study because we lack control data on how the chimpanzees would have behaved if they had not observed a model. However, we can make some inferences about social influences on learning. First, the chimpanzees in our study performed less-advanced manipulations in the pre-test situation before observing a model, and they showed continuous progress in the presence of a model. However, it is also true that they did not show evidence of immediate true imitation (Whiten & Ham, 1992). Their behaviour did not clearly improve immediately after observing successful nut cracking by a peer' (p.21) |
| Chimpanzees (*Pan troglodytes*) | Hopper et al. (2007) | Experimental study in captivity | One individual in the 'lift' seeded group spontaneously reinnovated the 'poke' methodology. | 'Poke was also not discovered by control animals tested individually. However, Poke emerged spontaneously in the Lift group and became dominant in both groups, regardless of the founder's Lift or Poke technique. Accordingly, this study demonstrated a statistically significant, differential spread of alternative techniques through social learning, yet no clear separation of traditions, unlike an earlier study with a different population of chimpanzees. This difference may be attributable to prior experience with relevant tools. In further experiments we investigated the basis of the social learning evident in acquisition of the Lift technique, using 'ghost' conditions in which the task was operated automatically rather than by a chimpanzee. Differential movement of the feeding device either by itself or with the tool coupled to it was not sufficient for learning to occur. It appears necessary for a chimpanzee to observe another chimpanzee performing the Lift technique for transmission to ensue' (p.1021) |

| Species | Reference | Testing methodology | Findings | Authors' interpretation |
|---|---|---|---|---|
| Chimpanzees (*Pan troglodytes*) | *Kitahara-Frisch & Norikoshi (1982)* | Experimental study in captivity | Two chimpanzees spontaneously used the leafy branches to retrieve the juice from the apparatus | 'The sponging behaviour as practised by zoo chimpanzees indicates that the example of the mother is by no means necessary for the habit to appear in young animals. This observation raises the question whether the acquisition of so-called proto-cultural habits does not rely as much, at least, on independent reinvention as on transmission through imitation learning. It is concluded that the sponge-making behaviour observed in Gombe can most parsimoniously be interpreted as an incidental corollary of a highly variable and potentially meaningful expression of the chimpanzee's behavioural resourcefulness' (p.41) |
| Chimpanzees (*Pan troglodytes*) | *Köhler (1925)* | Experimental study in captivity | Most of the tested chimpanzees spontaneously used the tools to rake in or reach the food | 'It is most difficult for chimpanzees to imitate anything, unless they themselves understand it (p.157)…when it (imitation) does occur, the situation, as well as its solution, must lie just about within the bounds set for spontaneous solution' (p.222) |
| Chimpanzees (*Pan troglodytes*) | *Kummer & Goodall (1985)* | Observational study in the wild | Report of chimpanzees at Gombe spontaneously using sticks to lever open boxes that contained bananas | 'One new behaviour that did spread through the community at Gombe was the use of sticks as levers to try to open banana boxes. Four and a half months after these boxes had been installed three adolescents began, independently, to use sticks to try to prize open the steel lids. Because a box was sometimes opened when a chimpanzee was working at it, the tool use was occasionally rewarded and, over the next year, the habit spread until almost all members of the community, including adult males, were seen using sticks in this way. That many individuals learnt as a result of watching their companions is suggested by the fact that one female was observed to behave thus on her very first visit to camp: before this she had had ample opportunity to watch what was going on from the surrounding vegetation (*Goodall, 1986*)' (p.212) |
| Chimpanzees (*Pan troglodytes*) | *Manrique & Call (2011)* | Experimental study in captivity | In experiment 1, four orangutans and one chimpanzee invented the use of a piece of electric cable to get the juice. Experiment 2 investigated whether subjects could transform a non-functional hose into a functional one by removing blockages that impeded the free flow of juice. Orangutans outperformed chimpanzees and bonobos by differentially removing those blockages that prevented the flow of juice, often doing so before attempting to extract the juice | 'In conclusion, orangutans proved to be more innovative than bonobos and most chimpanzees by inventing the use of tools as straws and modifying non-functional tools when needed. Moreover, some orangutans introduced the required modification prior to using the tools and were able to select novel suitable tools without having to try them first' (p.225) |
| Chimpanzees (*Pan troglodytes*) | *Menzel, Davenport & Rogers (1970)* | Experimental study in captivity | All the subjects spontaneously used sticks to rake in the bananas | 'The present data clearly show that early experience of a very general sort, before the age at which wild chimpanzees generally ordinarily become proficient at instrumental activities with objects, is required before animals can fully profit from later opportunities' (p.281) |

(Continued)

| Species | Reference | Testing methodology | Findings | Authors' interpretation |
|---|---|---|---|---|
| Chimpanzees (*Pan troglodytes*) | *Morimura (2003)* | Experimental study in captivity | All the subjects spontaneously used sticks to retrieve the juice | 'This finding demonstrated that when chimpanzees have the option to access juice through a variety of methods, they employ all available choices. It also supported the hypothesis that the behaviour of captive chimpanzees may come to resemble that of their wild counterparts as a function of behavioural freedom' (p.241) |
| Chimpanzees (*Pan troglodytes*) | *Motes-Rodrigo et al. (2019)* | Experimental study in captivity | All of the chimpanzees spontaneously used sticks to dig for the buried food | No interpretation offered by the authors |
| Chimpanzees (*Pan troglodytes*) | *Nash (1982)* | Experimental study in captivity | All the subjects used sticks to dip into the termite mound to retrieve the baited food | 'The artificial mound proved to be a viable simulation of the naturally occurring mounds, with most of the chimpanzees exploiting the food in the mound by using tools over the period of study. Interesting individual differences emerged in the way that the chimpanzees selected and used tools, some preferring to move some distance from the mound to collect "off-the-peg" tools. others preferring to sit and fashion a tool from material available nearer the mound. Also, some chimpanzees used both ends of a tool, while others used only one end' (p.211) |
| Chimpanzees (*Pan troglodytes*) | *Price et al. (2009)* | Experimental study in captivity | Two subjects were observed as being successful combiners and two twist-extenders in control condition | 'Social learning, therefore, had a powerful effect in instilling a marked persistence in the use of a complex technique at the cost of efficiency, inhibiting insightful tool use' (p.3377) |
| Chimpanzees (*Pan troglodytes*) | *Tomasello et al. (1987)* | Experimental study in captivity | One individual in the control group used the tool as a rake | 'None of the subjects demonstrated an ability to imitatively copy the demonstrator's precise behavioural strategies. More than simple stimulus enhancement was involved, however, since both groups manipulated the T-bar, but only experimental subjects used it in its function as a tool. Our findings complement naturalistic observations in suggesting that chimpanzee tool-use is in some sense 'culturally transmitted'—though perhaps not in the same sense as social-conventional behaviours for which precise copying of conspecifics is crucial' (p.175) |
| Chimpanzees (*Pan troglodytes*) | *Vale et al. (2017)* | Experimental study in captivity | The chimpanzees in the asocial control group, in which the behaviours were not seeded eventually individually acquired all of the methods to retrieve the bait, other than the 'unscrew' and 'suck' behaviours. | 'Five chimpanzees tested individually with no social information, but with experience of simple unmodified tool use, invented part, but not all, of the behavioural sequence. Our findings indicate that (i) social learning facilitated the propagation of the model-demonstrated tool modification technique, (ii) experience with simple tool behaviours may facilitate individual discovery of more complex tool manipulations, and (iii) a subset of individuals were capable of learning relatively complex behaviours either by learning asocially and socially or by repeated invention over time. That chimpanzees learn increasingly complex behaviours through social and asocial learning suggests that humans' extraordinary ability to do so was built on such prior foundations (p.635)… 'Thus, our captive populations, much like what has been documented in other captive populations (Hopper et al., 2014), were capable of re-inventing means in which their wild counterparts make and use tools.' As some individuals in one of the non-seeded groups eventually discovered the full behaviour, we cannot be sure that discovering it is outside the innovation capabilities of at least some chimpanzees (referred to by *Tennie, Call & Tomasello (2009)* as their 'zone of latent solutions')' (p.642) |

| Species | Reference | Testing methodology | Findings | Authors' interpretation |
|---|---|---|---|---|
| Chimpanzees (*Pan troglodytes*) | *Watson et al. (2017)* | Experimental study in captivity | One chimpanzee innovated a new solution: to slide the door up far enough to reach reward but not to lock mechanism, allowing her to then slide it down and retrieve a second reward | 'While this study has shown that chimpanzees are motivated to learn novel methods of accessing a resource from subordinate individuals, it is possible this is not true of forms of imitative behaviour that are thought to be normatively motivated and therefore, perhaps particularly directed toward important social partners' (p.23) |
| Chimpanzees (*Pan troglodytes*) | *Whiten, Horner & De Waal (2005)* | Experimental study in captivity | At least four chimpanzees in the seeded groups developed the alternative method to access the pan-pipes apparatus | 'Additionally, the 'two alternatives' methodology shows that learning involves not merely the facilitation of an existing competence, but a capacity to acquire particular local variants of the technique, precisely as required if the behavioural variants identified in wild populations are indeed socially transmitted' (p.738) |
| Chimpanzees (*Pan troglodytes*), bonobos (*Pan paniscus*) and orangutans (*Pongo pygmaeus abelii*) | *Visalberghi, Fragaszy & Savage-Rumbaugh (1995)* | Experimental study in captivity | All the tested species spontaneously solved both the simple and complex forms of the tasks, and inserted sticks into the tubes to retrieve the rewards | 'In conclusion, apes and capuchins can achieve success in a tool-using task and still have an apparently limited understanding of the causal relations involved (*Visalberghi & Limongelli, 1994*). This is also apparently true for human children (E. Visalberghi and A. Troise, 1994, unpublished data). We believe that successful use of an object as a tool to probe, rake, pound, or contain (the kinds of experimental tool-using tasks usually presented to non human primates) can be achieved through frequent spontaneous combinations of actions and objects, particularly objects with other objects or surfaces (*Fragaszy & Adams-Curtis, 1991*; *Schiller, 1952*). It appears that the experience of using tools does not, by itself, lead to or require the emergence of additional conceptual complexity' (p.59) |
| Chimpanzees (*Pan troglodytes*), gorillas (*Gorilla gorilla gorilla*), and orangutans (*Pongo pygmaeus abelii*) | *Hanus et al. (2011)* | Experimental study in captivity | Five individuals spontaneously developed the same solution of spitting into the tube to make the peanut float to the surface | 'According to the latency data (e.g. appearance of the first spit) the extra information of water inside the tube (wet condition) did not seem to stimulate chimpanzees' inventiveness. Furthermore, control tests showed that successful chimpanzees preferentially added water to the tube when the peanut was inside the tube, not simply when the peanut was present yet out of reach. Chimpanzees seemed to add water exclusively to affect the position of the peanut, which confirms the goal-directedness of their behaviour. Results are also consistent with the notion of insightful behaviour' (p.8) |
| Gorilla (*Gorilla gorilla gorilla*) | *Boysen et al. (1999)* | Experimental study in captivity | All the tested gorillas spontaneously used stick tools to fish peanut butter out from the artificial dome | 'Currently there is no compelling evidence to suggest that great apes are capable of reproducing a model's novel actions to obtain a goal in problem-solving situations (i.e. imitative learning)' (p.338) |
| Gorilla (*Gorilla gorilla gorilla*) | *Breuer, Ndoundou-Hockemba & Fishlock (2005)* | Observational study in the wild | An adult female gorilla was observed using a branch as a walking stick to test the water deepness and to aid in her attempt to cross a pool of water. In the second case, another adult female was observed using a detached trunk from a small shrub as a stabiliser during food processing. She then used the trunk as a self-made bridge to cross a deep patch of swamp | 'The observed tool use involved gorillas from two different groups and thus could indicate independent inventions, perhaps reflecting past negative experiences with deep water' (p.2042) |

| | Table 2 (continued) | | | |
|---|---|---|---|---|
| Species | Reference | Testing methodology | Findings | Authors' interpretation |
| Gorilla (*Gorilla gorilla gorilla*) | *Fontaine, Moisson & Wickings (1995)* | Observational study in captivity | Captive gorillas spontaneously used tools to rake in objects outside of their enclosure, as weapons, to sponge liquids and as ladders | 'The gorillas were not intentionally taught any of the skills described here. Although, in themselves the tasks accomplished with the materials at hand were relatively simple, their diversity is remarkable and indicative of certain innate cognitive skills rather than behaviour learnt by observing human activity' (p.223) |
| Gorilla (*Gorilla gorilla gorilla*) | *Nakamichi (1999)* | Observational study in captivity | Three captive gorillas (one female and two males) threw sticks into the foliage of trees, which the gorillas could not climb due to electric wire, to knock down leaves and seeds | 'This behaviour might originate from the nest-building behaviour which is commonly performed by gorillas in the wild' (p.494) |
| Gorilla (*Gorilla gorilla gorilla*) | *Natale, Poti' & Spinozzi (1988)* | Experimental study in captivity | Both the gorilla and the macaque spontaneously used the sticks provided to rake in the out-of-reach food | 'This difficulty in dealing with certain physical constraints is consistent with the hypothesis of a general lag in the development of physical cognition of non-human primates, which was suggested in previous studies of early sensorimotor development' (p.415) |
| Gorilla (*Gorilla gorilla gorilla*) | *Pouydebat et al. (2005)* | Experimental study in captivity | Every subject was able to make and use tools to extract food spontaneously. They carried branches to the apparatus, then broke them and removed projections such as leaves and bark | 'Our study attests to the fact that western lowland gorillas are able to manufacture and use food-extracting tools in a chimpanzee-like manner. The gorillas were able to adapt tools to a particular use by selecting branches and adapting their length to the depth of the holes. They may also anticipate the use of the tool, by beginning with the largest stick and progressively shortening it' (p.182) |
| Gorilla (*Gorilla gorilla gorilla*) | *Wood (1984)* | Observational study in captivity | Spontaneous stick tool use by gorillas | No interpretation offered by the authors |
| Gorillas (*Gorilla beringei beringei*) | *Kinani & Zimmerman (2015)* | Observational study in the wild | A young gorilla first tried to retrieve ants from an ant hole with her arm, but soon after selected a piece of wood and proceeded to insert the stick into the hole, withdraw the stick, and then lick ants off of the stick | 'This is the first time tool use has been reported in a wild mountain gorilla despite the intensive monitoring of this subspecies. The described tool use event is characterised as idiosyncratic and can, in part, be explained by Lisanga's curious nature as she is known to have an investigative personality. Furthermore, *Leca, Gunst & Huffman (2010)* mentioned that, despite the numerous examples of socially-transmitted tool use innovations in several non-human primate species, it should be noted that only a subset of such innovations become tradition and a large part of whether this happens likely depends on cost-benefit considerations. For wild mountain gorillas, ants are not a significant part of their diet perhaps because they do not offer substantial nutritional value per mass ingested, because other food sources are readily available, and/or because they are difficult to obtain without getting bitten' (p.355) |
| Gorillas (*Gorilla gorilla gorilla*) | *Neadle, Allritz & Tennie (2017)* | Experimental study in captivity | All subjects showed evidence of cleaning the apples in 75% of the trials | 'Given this occurrence of food cleaning in a culturally unconnected population of gorillas, we conclude that social learning is unlikely to play a central role in the emergence of the food cleaning behavioural form in Western lowland gorillas; instead, placing a greater emphasis on individual learning of food cleaning's behavioural form' (p.1) |

| Species | Reference | Testing methodology | Findings | Authors' interpretation |
|---|---|---|---|---|
| Gorillas (*Gorilla gorilla gorilla*), chimpanzees (*Pan troglodytes*) | Lonsdorf et al. (2009) | Experimental study in captivity | All chimpanzees except for the alpha male attempted to fish on the very first trial. Some gorillas were successful on the first day and some individuals never attempted the task even after 60 bait trials | 'Together, these results suggest that chimpanzees may be better equipped to acquire knowledge that is socially transmitted. Whether individual or social learning is necessary to learn the task, access to the device is critical, and our observations suggest that, in gorillas, subordinate individuals may be socially constrained from accessing the termite mound' (p.1119) |
| Orangutan (*Pongo pygmaeus abelii*) | Mendes, Hanus & Call (2007) | Experimental study in captivity | All the orangutans collected water from a drinker and spat it inside the tube to get access to the peanut | 'The sudden acquisition of the behaviour, the timing of the actions and the differences with the control conditions make this behaviour a likely candidate for insightful problem solving' (p.453) |
| Orangutan (*Pongo pygmaeus abelii*) | Bandini et al. (2020a) | Experimental study in captivity | One juvenile orangutan spontaneously started cracking nuts using the tools provided | No interpretation offered by the authors |
| Orangutan (*Pongo pygmaeus abelii*) | Lehner, Burkart & Van Schaik (2011) | Experimental study in captivity | Naive captive orangutans first spontaneously started using sticks to retrieve liquids from a testing apparatus, and then built upon this behaviour to create more efficient techniques when the simple branch-dipping method was rendered inefficient | 'Most subjects started with a nonefficient technique ("dip stick"), but then spontaneously came up with innovative and more efficient solutions, and eventually largely switched to these and preferentially used them. Minimally, this indicates that they recognised which of two techniques yields a higher return. At the same time, most individuals kept on using a variety of techniques under the regular condition, predicting they would show high flexibility to abandon preferred techniques and switch to different techniques as the condition was changed' (p.453) |
| Orangutan (*Pongo pygmaeus abelii*) | Nakamichi (2004) | Experimental study in captivity | Three adult orangutans spontaneously stripped leaves and twigs from a branch provided and then inserted the tool into a hole to obtain the baited food. When the orangutans were unable to insert a tool into a hole, they modified the tool and/or changed their tool-using technique, such as changing how they grasped the tool | 'Analysing the spontaneous use of sticks as tools by gorillas in captivity can lead to a better understanding of not only their cognitive ability but also of their social relationships, which may otherwise be concealed' (p.487) |
| Monkeys | | | | |
| Baboons (*Papio anubis*) | Benhar & Samuel (1978) | Experimental study in captivity | Both of the tested baboons used the metal hooks provided to pull up the bread | 'The successful use of a tool by two baboons in our colony indicates that "specific training" was not necessary, since these animals had previously used their tails for retrieval. These results support the statement by Hall (1963) that baboons have so far not been seen to demonstrate tool-using in the wild for food seeking behaviour, but they do so readily when given the opportunity in captivity' (p.388) |
| Baboons (*Papio cynocephalus anubis*) | Westergaard (1992) | Experimental study in captivity | The baboons used paper, browse, and other materials as tools to extract sweet liquids from the apparatus | 'Results demonstrate flexible combinatorial manipulation and spontaneous use of tools by infant baboons. These data are consistent with hypotheses that (1) an evolutionary history of omnivorous extractive foraging is associated with the use of tools and (2) free play in an object-enriched captive environment may facilitate combinatorial manipulation in nonhuman primates' (p.1) |

(Continued)

| Species | Reference | Testing methodology | Findings | Authors' interpretation |
|---|---|---|---|---|
| Baboons (*Papio papio*) | Beck (1973) | Experimental study in captivity | A sub adult male spontaneously used a tool to rake in food. Only one other individual demonstrated the same behaviour after a long period in which only the first male was using a tool as a rake | 'A sub adult male from a captive group of Guinea baboons learned, by trial-and-error, to use a tool to rake in food. He then used the tool 104 times over 26 days, thereby providing the group with most of its food. No other group member used the tool during this period. The tool user was removed, and the remainder of the group was given access to the display. None imitated his tool use. It took longer for another finally to learn to use the tool than it had for the initial solution. However, compared with the period before initial solution, group members manipulated the tool more frequently and touched the food pan with the tool nearly twice as many times after the tool user's separation. This type of tool use appears to be too complex for baboons to imitate directly. However, as a result of observing successful tool use, they attend more to the problem and manipulate the tool more frequently and more accurately. This increase in frequency and accuracy may, in turn, accelerate acquisition of the response by observers through instrumental trial-and-error learning' (p.579) |
| Baboons (*Papio papio*) | Petit et al. (1993) | Observational study in captivity | Baboons were found to spontaneously use stones in a pounding action to enlarge a hole in their outdoor enclosure (the purpose behind this behaviour is unknown) | 'Given the timing of events, it can be inferred that Voy was the initial discoverer and that the acquisition by the other individuals was facilitated by his behaviour…As in other cases of social influence of learning, stimulus enhancement probably played an important part in the dissemination of the behaviour' (p. 163) |
| Baboons (*Papio papio*) | Westergaard & Fragaszy (1987) | Experimental study in captivity | Although the macaques did not use any of the materials provided to retrieve the syrup, they did use the sticks and other materials to make ladders to climb to the top of their enclosure | 'A variety of goal-directed manipulative activities (use of objects to act as ladders, to apply leverage, and to create perches) occurred spontaneously, with some instances involving joint action or social use. These data are consistent with the hypotheses that macaques possess extensive capacities for object exploration and social facilitation, and that an evolutionary history of omnivorous foraging habits correlates positively with the expression of anomalous sensorimotor skills' (p.231) |
| Black-capped capuchins (*Sapajus apella*) | Izawa & Mizuno (1976) | Observational study in the wild | On a border of La Macarena National Park in Colombia, the authors observed the feeding behaviour of a black-capped capuchin, in which the monkey fed on the albumen of the fruit of cumare, a kind of cocoid palm-fruit, using two different methods, according to the degree of ripeness of the fruit | 'The characteristic behaviour developed by the black-capped capuchin while eating the fruit of cumare could be fixed as one of the higher level adaptive behaviours of the animal to his habitat' (p.1) |
| Capuchins (*Cebus apella*) | Antinucci & Visalberghi (1986) | Experimental study in captivity | One capuchin spontaneously started cracking nuts with tools when provided with the materials. Always picked the right tool out of several options (stone, wood plastic) | 'C's results in the tool condition show that this monkey is capable of appropriately utilising any detached object offered to it as a hammer tool, even if it has no previous experience with it, and that, furthermore, when given a choice, it is also capable of selecting the most efficient object from those available' (p.361) |
| Capuchins (*Cebus apella*) | Visalberghi & Trinca (1989) | Experimental study in captivity | Three out of four capuchins spontaneously used sticks to push food out of a tube. They were also successful in modifying the stick when needed | 'The first success of the adult male has all the characteristics of an insight solution' (p.519) |

| Species | Reference | Testing methodology | Findings | Authors' interpretation |
|---|---|---|---|---|
| Capuchins (*Cebus libidinosus*) | *Waga et al. (2006)* | Observational study in the wild | Spontaneous emergence of stone tool-use in a group of wild capuchins. The capuchins were observed using stones to crack open hard-shelled Jatoba fruit (*Hymenaea courbaril*) | 'We propose that the probability of the emergence of the use of pounding stones as tools may be dependent on the ecological variables that influence the degree of terrestriality and extractive foraging and the complex interaction of these factors' (p.337) |
| Capuchins (*Cebus nigritus*) | *Bortolini & Bicca-Marques (2007)* | Observational study in captvitiy | An adult female was observed banging a twig with a piece of stone against a larger stone, and then licking/chewing and likely extracting something from it with her mouth. She was then observed probing an unseen structure (probably a hole in the enclosure's drinking fountain) with the modified twig | 'Although we do not know what happened immediately prior to this behavioural sequence and could not see whether the female acquired anything as a result of probing, the speed at which this sequence of events occurred is highly suggestive of a causal understanding during object manipulation and seems to qualify as a case of spontaneous tool-making' (p.75) |
| Capuchins (*Sapajus apella*) | *Visalberghi (1987)* | Experimental study in captivity | Two capuchins started cracking nuts with blocks spontaneously | 'In the present study, two tufted capuchins spontaneously acquired a novel form of tool-use, consisting in cracking nuts with a block, which was not innate or stereotyped' (p.176) |
| Capuchins (*Sapajus apella*) | *Westergaard & Fragaszy (1985)* | Experimental study in captivity | The capuchins spontaneously started using cups as drinking tools and to carry objects around the enclosure. Also used paper-towels as sponges to soak up water | 'The repeated, spontaneous use of provided objects as tools in this group further illustrates the influence of environmental conditions on the form and frequency of activity in captive primate groups' (p.326) |
| Capuchins (*Sapajus apella*) | *Westergaard & Fragaszy (1987)* | Experimental study in captivity | Several of the subjects spontaneously manufactured and used tools to scoop water from their water bowls. Subsequently other objects that had been provided (such as tupperwares) were used to transport water, food, and other objects across the enclosure | 'Immature monkeys in this group developed tool-use more gradually than adults. However, subsequent to initial success, immature monkeys quickly became as proficient as adults that used tools' (p.327) |
| Capuchins (*Sapajus apella*) | *Westergaard & Suomi (1994a)* | Experimental study in captivity | Several of the naive capuchins spontaneously hit the hammerstone against the core to make sharp flakes. The flakes were then used to cut open the puzzle box | 'We suggest that the ability to make and use simple stone tools may have been discovered numerous times and utilised by more than one hominid genus and species' (p.403) |
| Gibbons (*Hylobatidae*) | *Beck (1966)* | Experimental study in captivity | Except for one individual, every other subject in the group was able to spontaneously solve all of the types of problems presented, and use string as a tool to retrieve the food | 'The data clearly show that gibbons are able to solve the type of problem used by Kôhler (1959) to demonstrate insight. With one exception (every presentation of the Type II problem to LAF was discontinued) every animal was able to solve all of the types of problems presented[…]. A case of tool use may indeed be an insightfully learned behavioural sequence but it also may be one shaped by the patient application of operant techniques (in the wild by a fortuitous chain of environmental events) or it may even be a sequence which is not learned at all but rather transmitted genetically to the animal; for example shrikes' using thorns on which to impale their prey' (p106) |

(Continued)

| Species | Reference | Testing methodology | Findings | Authors' interpretation |
|---|---|---|---|---|
| Gibbons (*Hylobatidae*) | *Geissmann (2009)* | Observational study in captivity | A captive female white-handed gibbon slammed the sliding door of her wooden sleeping box during the climax if a display bout | 'In summary, the tool use in a female gibbon reported in this paper presents a singularity, as in several other reports on gibbon cognitive abilities' (p.58) |
| Golden Lion Tamarins (*Leontopithecus rosalia rosalia*) | *Stoinski & Beck (2001)* | Observational study in captivity | Two types of tool use were observed in eight captive, free-ranging golden lion tamarins. All eight individuals used twigs and/or radio collar antennae to pry bark from trees and probe crevices, presumably for invertebrates. Three individuals used tools for grooming. Additionally, twigs were used as fishing tools in attempts to extract substances from tree holes or other crevices | 'Social transmission may be one of the mechanisms responsible for the acquisition of tool use—six of the eight tool users resided in two social groups, and the only two individuals that used antennae as grooming tools were a bonded pair' (p.319) |
| Japanese Macaques (*Macaca Fuscata*) | *Leca, Gunst & Huffman (2010)* | Observational study in the wild | The individual stretched one or a few pieces of its own hair or another individual's hair and inserted the hair between the upper or lower front teeth by performing repeated teeth-chattering to remove food remains stuck between the teeth. So far, this behaviour has only been observed in one individual, and has not spread to the rest of the group | 'Because chance may account for a good number of behavioural innovations (*Reader & Laland, 2003*), and DF was always associated with grooming activity, we suggest that the DF innovation is an accidental by-product of grooming' (p.19) |
| Japanese Macaques (*Macaca Fuscata*) | *Machida (1989)* | Observational study in captivity | A juvenile female spontaneously began standing poles against a concrete wall and climbing up them, in the next couple of years the rest of the group started showing the same behaviour | 'In a captive group of Japanese monkeys, a juvenile female spontaneously began standing poles against a concrete wall and climbing up them in 1983. By 1987, 3 juvenile females out of 39 monkeys had acquired the behaviour' (p.1) |
| Japanese Macaques (*Macaca fuscata*) | *Tokida et al. (1994)* | Experimental study in captivity | The macaques used the hook tool, and one individual spontaneously threw stones into the tube to push the apple out of the other side of the tube. The same individual then coaxed her infants to go into the tube and retrieve the food | 'Captive tufted capuchin monkeys, *Cebus apella*, learnt by trial and error within 2 h to use sticks to remove food from a transparent pipe, although Cebus seem to have a strong propensity for spontaneous tool use' (p.1025) |

**Table 2** (*continued*)

| Species | Reference | Testing methodology | Findings | Authors' interpretation |
|---|---|---|---|---|
| Lion-Tailed Macaques (*Macaca silenus*) | *Westergaard (1988)* | Experimental study in captivity | Several macaques spontaneously manufactured and used tools to extract syrup from the apparatus | 'This report is the first to describe spontaneous manufacture of tools in any group of Old World monkeys and provides evidence of greater continuity among primates for the expression of complex cognitive abilities. These data are consistent with hypotheses that lion-tailed macaques have extensive propensities for advanced sensorimotor skills and that omnivorous, extractive foraging is associated with the manufacture and use of tools' (p.152) |
| Lion-Tailed Macaques (*Macaca silenus*) | *Westergaard & Lindquist (1987)* | Experimental study in captivity | Four macaques in the group created ladders by propping long pieces of browse or bamboo poles against vertical cage structures and climbing them | 'Findings indicate richness in the frequency and form of manipulative activities, with juvenile males manipulating the test objects more frequently and exhibiting more goal-directed manipulative activity than adult females. A variety of goal-directed manipulative activities (use of objects to act as ladders, to apply leverage, and to create perches) occurred spontaneously, with some instances involving joint action or social use. These data are consistent with the hypotheses that macaques possess extensive capacities for object exploration and social facilitation, and that an evolutionary history of omnivorous foraging habits correlates positively with the expression of anomalous sensorimotor skills' (p.1) |
| Long-Tailed Macaques (*Macaca fascicularis fascicularis*) | *Watanabe, Urasopon & Malaivijitnond (2007)* | Observational study in the wild | One macaque was observed using human hair as dental floss | 'This behaviour could be considered a newly occurring cultural behaviour, which has become established under very specialised circumstances' (p.1) |
| Long-Tailed Macaques (*Macaca fascicularis fascicularis*) | *Zuberbühler et al. (1996)* | Experimental study in captivity | One macaque spontaneously used a stick to rake in fruit. After a couple of years, five more individuals showed the same behaviour | 'Since increased manipulations were only recorded at times when we caused tool-use behaviour in MD by presenting fruits, and since the manipulations were performed on the same object class MD was using as a tool, we conclude that MD's activity was the cause for these increased activities' (p.9) |
| Pig-tailed Macaques (*Macaca nemestrina*) | *Chiang (1967)* | Observational study in the wild | First observation of wild macaques using leaves to wash dirt/mud off seeds and fruit before eating them | No interpretation offered by the authors |
| Rhesus monkeys (*Macaca mulatta*) | *Parks & Novak (1999)* | Experimental study in captivity | Three females used a variety of cup-like containers as drinking utensils. All of the individuals used the trough to soak chow prior to ingestion | 'In our study, female rhesus monkeys used objects for scooping and carrying water in a fashion similar to that described for capuchins. Some of our subjects drank from objects in the same way as that described for an adult male orangutan (*Miller & Quiatt, 1983*)-by scooping water into a hollow object and then raising the object to the mouth and drinking from it' (p.22) |
| Tonkean Macaques (*Macaca tonkeana*) | *Anderson (1985)* | Experimental study in captivity | Two individuals spontaneously used the rod to retrieve the honey | 'The results of the present study provide further evidence for the ability of macaques to spontaneously acquire new behavioural sequences involving object manipulation to obtain a goal such as food which is unattainable directly[…]The failure of the other members to acquire the new behaviour can be interpreted in terms of the conservatism of adults and the fact that the two adolescents tended to monopolise working with the rod once they had knowledge of its value' (p.14) |

(Continued)

| Species | Reference | Testing methodology | Findings | Authors' interpretation |
|---|---|---|---|---|
| Tonkean Macaques (*Macaca tonkeana*) | *Ducoing & Thierry (2005)* | Experimental study in captivity | Three out of the five subjects spontaneously used the wooden pole to retrieve the banana. The other two subjects never showed the behaviour | 'In the first experiment, three males learned individually to obtain a food reward using a wooden pole as a climbing tool. They began using the pole to retrieve the reward only when they could alternatively experience acting on the object and reaching the target. In a second experiment, we first tested whether four other subjects could learn branch leaning after having observed a group-mate performing the task. Despite repeated opportunities to observe the demonstrator, they did not learn to use the pole as a tool. Hence we exposed the latter subjects to individual learning trials and they succeeded in the task' (p.103) |
| Tonkean Macaques (*Macaca tonkeana*) | *Ueno & Fujita (1998)* | Observational study in captivity | One subject used a withered soft stalk of a coconut leaf to obtain a piece of food on the ground out of his reach | No interpretation offered by the authors |
| Tufted Capuchins (*Cebus apella*) | *Fernandes (1991)* | Observational study in the wild | A wild capuchin was observed opening oyster shells fixed to the mangrove by hitting them rapidly and repeatedly with a hand-held object which was a piece of the oyster colony itself | '*C. apella* is by far the most widely-distributed platyrrhine species and the behavioural adaptability of these capuchins, which enables them to exploit alternative resources in habitats, like the mangrove swamp, where typical primate foods such as fruit are relatively scarce, is undoubtedly a key factor in the ecological success of the species' (p.530) |
| Tufted Capuchins (*Cebus apella*) | *Ottoni & Mannu (2001)* | Observational study in the wild | The group of capuchins was observed to start cracking nuts using two stones, one with a horizontal surface, laid on the substrate (the anvil) and the other, smaller, held in the hands (the hammer) | 'As with chimpanzees, infant capuchins are highly tolerated by adults, and episodes involving the observation of older, more proficient individuals by younger ones, sometimes followed by manipulation of the stones by the youngster, point to a possibly role for some kind of observational learning, albeit restricted to stimulus enhancement, as a starting point of a long process of individual improvement by trial-and-error' (p.357) |
| Tufted Capuchins (*Cebus apella*) | *Struhsaker & Leland (1977)* | Observational study in the wild | Three individuals were observed using stones to crack open encased palm nuts | 'The exploitation of this abundant food resource by *C. apella* and not by any of the other seven sympatric primate species may give them a competitive advantage. This circumstance could partially account for the numerical superiority of *C. apella* at the time of our survey' (p.1) |
| White-faced Capuchin (*Cebus capucinus*) | *Boinski (1988)* | Observational study in the wild | First observation of tool-use in white-faced capuchins. The monkey used a branch to repeatedly hit a venomous snake (*Bothrops asper*) | 'This is the first direct observation of the use of a tool by a wild capuchin. Given the abundant evidence of tool use by capuchins in captivity, this anecdote is not very surprising. In an environment where appropriate stimuli are abundant, snake-bashing might have been discovered by a chance variation of a branch-shaking or a branch-throwing display' (p.178) |
| Wild-bearded capuchins (*Sapajus libidinosus*) | *Proffitt et al. (2016)* | Observational study in the wild | Wild bearded capuchin monkeys in Brazil were observed deliberately break stones, unintentionally producing smaller fractured stones | 'The capuchin data add support to an on-going paradigm shift in our understanding of stone tool production and the uniqueness of hominin technology. Within the last decade, studies have shown that the use and intentional production of sharp-edged flakes is not necessarily tied to the genus Homo. Capuchin SoS percussion goes a step further, demonstrating that the production of archaeologically identifiable flakes and cores, as currently defined, is no longer unique to the human lineage' (p.3) |
| **Birds** | | | | |
| Canaries (*Serinus canaria*) | *Hinde & Warren (1959)* | Experimental study in captivity | All the canneries spontaneously made nests indistinguishable to those of their wild counterparts | 'Those birds which had material continuously, but were not permitted to construct a nest, built more actively than those allowed to build undisturbed. Birds without material for most of the time built vigorously during the watches with material, but seldom visited the nest-pan at other times. Birds without a nest-pan showed active building behaviour but it was mainly limited to the early phases of the nest-building sequence' (p.1) |

| Species | Reference | Testing methodology | Findings | Authors' interpretation |
|---|---|---|---|---|
| Goffin cockatoo (*Tanimbar corella*) | *Auersperg et al. (2012)* | Experimental study in captivity | One individual broke a large splinter off a beam outside of the enclosure, and then used the stick as a rake to retrieve the cashews and was successful in all 10 trials | 'Our observations prove that innovative tool-related problem-solving is within this species' cognitive resources. As it is unknown for tools to play a major role in this species' ecology, this strengthens the view that tool competences can originate on general physical intelligence, rather than just as problem-specific ecological solutions' (p.2) |
| Goffin cockatoo (*Tanimbar corella*) | *Auersperg et al. (2016)* | Experimental study in captivity | All the cockatoos were able to spontaneously make tools out of all the materials provided to retrieve the cashew nut | 'We show that an Indonesian generalist parrot, the Goffins cockatoo, can flexibly and spontaneously transfer the manufacture of stick-type tools across three different materials. Each material required different manipulation patterns, including substrates that required active sculpting for achieving a functional, elongated shape' (p.1) |
| Goffin cockatoo (*Tanimbar corella*) | *Laumer et al. (2017)* | Experimental study in captivity | The birds individually acquired the ability to bend hook tools from straight wire to retrieve food from vertical tubes and four subjects unbent wire to retrieve food from the horizontal tubes | 'Pre-experience with ready-made hooks had some effect but was not necessary for success. Our results indicate that the ability to represent and manufacture tools according to a current need does not require genetically hardwired behavioural routines, but can indeed arise innovatively from domain general cognitive processing' (p.1) |
| Hawaiian crows (*Corvus hawaiiensis*) | *Rutz et al. (2016)* | Experimental study in captivity | All the subjects in the group reliably and spontaneously used tools in all the extraction tasks provided | 'This indicates that, despite considerable social mixing, it is unlikely that a single 'innovation' event can explain the observed species-wide distribution of tool competence. 'Alalā clearly possess a propensity to 'discover' tool-assisted foraging solutions independently, which probably results from genetically canalised, persistent object-exploration behaviour' (p.404) |
| Hyacinth macaws (*Anodorhynchus hyacinthinus*) | *Borsari & Ottoni (2005)* | Observational study in captivity | Six captive macaws were observed using wood as a lever to crack open nuts in their enclosure, similar to the behaviour they show in the wild | 'Our data suggest that hyacinth macaws have an innate tendency to place objects (tools) and food together inside their beaks, while feeding. Four of our six subjects were captive bred, separated from their parents at hatching time and hand-raised, so they did not have the opportunity to watch a more experienced adult performing this behaviour. On the other hand, we cannot rule out the possibility that the younger juveniles could have learned to use tools from watching the older ones (nor that social learning could play a greater role in more naturalistic settings)' (p.51) |
| New Caledonian Crow (*Corvus moneduloides*) | *Hunt (1996)* | Observational study in the wild | One of the first reports of hook tool-making of wild New Caledonian crows | 'Crows have achieved a considerable technical capability in their tool manufacture and use' (p.249) |
| New Caledonian Crow (*Corvus moneduloides*) | *Kenward et al. (2005)* | Experimental study in captivity | All the juvenile New Caledonian crows spontaneously manufactured and used tools, including the subjects that never had contact with adults of their species or any prior demonstration by humans | 'In the light of our findings, it is possible that the high level of skill observed in wild adult crows is not socially acquired. Social input, however, may be important in transmitting specific techniques and tool shapes. This idea is supported by the close attention our juveniles paid to demonstrations of tool use by their human foster parents' <br><br> 'The fact that an inherited predisposition can account for a complex behaviour such as tool manufacture highlights the need for controlled investigation into behavioural ontogeny in other species that seemingly show culturally transmitted behaviour' (p.121) |

(Continued)

| Species | Reference | Testing methodology | Findings | Authors' interpretation |
|---|---|---|---|---|
| New Caledonian Crow (*Corvus moneduloides*) | *Taylor et al. (2007)* | Experimental study in captivity | On their first attempt to solve the problem, six out of the seven naive crows used the short tool to probe the tool-box with the long tool, and then used the long tool to retrieve the baited food | 'The experiments revealed that the crows did not solve the metatool task by trial-and-error learning during the task or through a previously learned rule. The sophisticated physical cognition shown appears to have been based on analogical reasoning. The ability to reason analogically may explain the exceptional tool-manufacturing skills of New Caledonian crows' (p.1504) |
| New Caledonian Crow (*Corvus moneduloides*) | *Taylor et al. (2010)* | Experimental study in captivity | All the crows spontaneously solved the first task (pulling the string), however not all the subjects solved the task in the second experiment, and performance only increased with exposure | 'However, when visual feedback was available via a mirror mounted next to the apparatus, two naïve crows were able to perform at the same level as the experienced group. Our results raise the possibility that spontaneous string pulling in New Caledonian crows may not be based on insight but on operant conditioning mediated by a perceptual-motor feedback cycle' (p.1) |
| New Caledonian Crow (*Corvus moneduloides*) | *Von Bayern et al. (2009)* | Experimental study in captivity | Two subjects picked up stones and dropped them into the tube, despite having never used stones as tools or seen stones being dropped into the apparatus. The remaining two stone-naive subjects did not solve the task in the permitted time. However, one of them successfully used stones in a retest after having spontaneously pushed the platform with sticks in a stick/stone choice experiment | 'Our results show that, for New Caledonian crows, learning about some functional affordances of the task (collapsibility of the platform through force or contact) is essential, whereas learning about specific visual stimuli (stones acting on the platform) or actions (picking up and dropping stones) is not' (p.1) |
| New Caledonian Crow (*Corvus moneduloides*) | *Weir (2002)* | Experimental study in captivity | One female bent the wire to make a hook to retrieve the food in nine out of the ten valid trials | 'Thus, at least one of our birds is capable of novel tool modification for a specific task. In the wild, New Caledonian crows make at least two sorts of hook tools using distinct techniques but the method used by our female crow is everyday physics (from their manipulative experience), but she had no model to imitate and, to our knowledge, no opportunity for hook-making to emerge by chance shaping or reinforcement of randomly generated behaviour' (p.981) |
| New Caledonian Crow (*Corvus moneduloides*) and Kea parrots (*Nestor notabilis*) | *Auersperg et al. (2011)* | Experimental study in captivity | At least one individual of each species discovered all four available methods | 'At least one individual of each species discovered all four available methods. This proves that in principle, the affordances of the tasks lay within the cognitive and physical capacity of both species' (p.5) |
| Northern Blue Jays (*Cyanocitta cristata*) | *Jones & Kamil (1973)* | Observational study in captivity | Six captive blue-jays spontaneously crumpled newspaper that was in the enclosure to rake in pellets that were just outside of the enclosure | 'Although a definitive answer is not possible, we feel that the behaviour was first acquired serendipitously by a single blue jay [...]The fact that to date we have found six jays in our colony demonstrating tool-using behaviour is thought to be more likely the result of the spread of the behaviour through observational learning or imitation than the result of the independent acquisition of this behaviour by each of the six jays' (p.1087) |
| Pigeon (*Columba livia domestica*) | *Epstein (1987)* | Experimental study in captivity | All of the subjects pushed the provided box under the banana to retrieve it | 'Here we appear to have on hand an instance of insightful problem solving' (p.62) |

| Species | Reference | Testing methodology | Findings | Authors' interpretation |
|---|---|---|---|---|
| Rooks (*Corvus frugilegus*) | Bird & Emery (2009a) | Experimental study in captivity | All the naive rooks made, used and modified different tools (stones, sticks, hooks) spontaneously to retrieve the worm | 'The initial solution to the task may have been derived from the birds' prior experience. Seemingly insightful behaviours may be achieved by "chaining" previously rewarded behaviours or by generalising from one task to another. Although the subjects' behaviour can be explained in these ways, there is some reason to suggest that the behaviour was not solely a conditioned action: multiple acts of stone dropping were necessary for success (in previous experiments, one stone had been necessary for success), and subjects did not try to reach for the reward after dropping each stone. In addition, they reached for the worm from the top of the tube rather than checking at the base (in previous experiments, the worm was accessible below the tube)' (p.1411) |
| Rooks (*Corvus frugilegus*) | Bird & Emery (2009b) | Experimental study in captivity | All four subjects solved the task and spontaneously started using stones to raise the water level to a height at which the worm could be reached | 'It is possible that the initial stone-dropping behaviour was elicited by subjects' previous experience and that the increased proximity of the worm reinforced the initial stone drop, leading to a cycle of stone dropping until the worm could be reached. However, it is not clear that the worm getting closer would seem rewarding to the subject. Rather, the worm getting closer but still not being within reach might equally have been unrewarding or frustrating; hence, the behaviour would be repeated only if it were goal directed' (p.1141) |
| Rooks (*Corvus frugilegus*) | Seed et al. (2006) | Experimental study in captivity | All the rooks spontaneously used sticks in the trap-tube task to pull the food towards them | 'One possibility for this is the fact that, although they are not tool-users, rooks are the only corvids reported to cache food by digging a hole before placing the food inside it and then covering it over. Given that the traps are effectively holes, it may be that learning what constitutes a functional hole is an ecologically relevant problem for a rook. Corvids are also opportunistic generalists for whom rapid learning is likely to confer a high survival advantage. Other possible explanations for this difference may be considered: Shettleworth has pointed out that many of the primates used in similar experiments have participated in a variety of other physical tasks (unlike our rooks), which could have interfered with their learning; secondly, the rooks were not required to insert a tool, and this could also potentially facilitate learning on the task; lastly, it is possible that pulling food toward oneself is a more natural behaviour for animals than pushing it away, and this improvement in "external validity" may also be a facilitatory factor' (p.700) |
| Rooks (*Corvus frugilegus*) | Tebbich et al. (2006) | Experimental study in captivity | All the rooks used the stick provided in the apparatus to retrieve the food | 'This study demonstrates that rooks, which do not spontaneously use tools, can solve the trap-tube problem. Our findings are in line with those of Hauser and colleagues (Hauser, Krali & Botto-Mahan, 1999; Hauser, Pearson & Seelig, 2002), who have found that non-tool-using primates are capable of learning the solution to other physical problems, such as those based on connectedness' (p.229) |
| Woodpecker finch (*Cactospiza pallida*) | Tebbich & Bshary (2004) | Experimental study in captivity | One of six woodpecker finches was able to solve the trap tube task, and several individuals modified tools and chose twigs of appropriate length | 'Tool use in the woodpecker finch is not a stereotypic behavioural pattern, but is open to modification by learning[…]Tool use in the woodpecker finch also seems to be guided by a rapid process of trial and error learning' (p.8) |

(Continued)

| Species | Reference | Testing methodology | Findings | Authors' interpretation |
|---|---|---|---|---|
| Woodpecker finch (*Cactospiza pallida*) | *Tebbich et al. (2001)* | Experimental study in captivity | One individual in the no-model group started using tools during testing conditions | 'However, we found that not all adult woodpecker finches used tools in our experiments. These non-tool-using individuals also did not learn this task by observing tool-using conspecifics. Our results suggest that tool-use behaviour depends on a very specific learning disposition that involves trial-and-error learning during a sensitive phase early in on in ontogeny' (p.2189) |
| **Other Animals** | | | | |
| Asian Elephant (*Elephas maximus*) | *Chevalier-Skolnikoff & Liska (1993)* | Experimental study in captivity | 21 different types of spontaneous tool-use were observed in the captive groups of elephants, when the elephants were provided with more tools in their enclosure. Most behaviours involved using tools (such as sticks and leaves) for body-care purposes | 'Tool use in elephants occurs mainly in the contexts of body care, parasite control and body cooling, and may reflect an adaptive function in these nearly furless land mammals' (p.217) |
| Asian Elephant (*Elephas maximus*) | *Foerder et al. (2011)* | Experimental study in captivity | One elephant showed spontaneous problem solving by moving a large plastic cube, on which he then stood, to acquire the food. In further testing he showed behavioural flexibility, using this technique to reach other items and retrieving the cube from various locations to use as a tool to acquire food | 'These results provide experimental evidence that an elephant is capable of insightful problem solving through tool use. Evidence for this ability is indicated by the suddenness of Kandula's problem solving behaviour without evidence of prior trial and error learning' (p.3) |
| Bear (*Ursos arctos*) | *Deecke (2012)* | Observational study in the wild | One bear was observed picking up barnacle-encrusted rocks in shallow water, manipulating them, re-orienting them in its forepaws, and then using them to rub its neck and muzzle | 'However, social learning may not be necessary to explain the spread of stone-rubbing even if this form of tool-use was found to be common: brown bears frequently turn over rocks in search of food and feed on intertidal barnacles…both of which would provide ample opportunity for the acquisition of stone-rubbing behaviour through individual learning alone' (p.7) |
| Bear (*Ursos arctos*) | *Waroff et al. (2017)* | Experimental study in captivity | Six out of the eight tested bears manipulated and used objects to climb on top of to reach suspended food | 'Our observations in captive bears reveal that bears very often use physical force when approaching new problems which will lead to trial and error problem-solving…When force does not work, bears often appear to demonstrate insight like behaviour, with individual variability' (p. 63) |
| Dingo (*Canis dingo*) | *Smith, Appleby & Litchfield (2011)* | Observational study in captivity | A captive male dingo spontaneously learned to move objects around his enclosure, apparently to multiple ends, such as in an effort to gain the additional height required to attain objects otherwise out of reach, or to attain a better view of his surroundings | 'These behaviours may have emerged as a result of simple learning processes and reinforcement rather than by forming an 'insightful solution' (as suggested by Köhler for chimpanzees who displayed similar behaviours, 1927/1971; for criticisms of insight learning see *Windholz & Lamal, 1985*). It is therefore possible that dingoes, like other wild canids (e.g. coyotes, *Coppinger & Coppinger, 2001*; wolves, *Mech, 1991*) appear to be able to replicate behaviours performed by sanctuary staff as a result of observational learning (e.g. wolves, *Frank, 1980*). Through trial-and-error (or 'trial-and-success') learning (*Thorndike, 1898*; *Chance, 1999*) or observational learning (*Bandura, 1965*) with Sanctuary staff serving as observable models by moving objects during routine enclosure maintenance, Sterling may have formed complex associations and applied them to different contexts (escape, view, or attain objects out-of-reach)' (p.223) |

| Species | Reference | Testing methodology | Findings | Authors' interpretation |
|---------|-----------|---------------------|----------|-------------------------|
| Octopus (*Amphioctopus marginatus*) | *Finn, Tregenza & Norman (2009)* | Observational study in the wild | Wild octopus was observed carrying around two coconut halves that were then used as a defensive protective shell when needed | 'The behaviour reported here is likely to have evolved using large empty bivalve shells prior to the relatively recent supply of the clean and light coconut shell halves discarded by the coastal human communities adjacent to the marine habitat of this species' (p.2) |
| Sea Otter (*Enhydra lutris nereis*) | *Nicholson et al. (2007)* | Observational study in captivity | Orphaned sea otters in a research facility spontaneously developed rudimentary pound hammering behaviours to open encased food | 'Pounding open clams was positively correlated with biting open mussels, so rehabilitation methods had a similar effect on the development of this milestone. Surrogate-reared pups, therefore, developed all foraging skills at a significantly younger age (by 2–3 weeks) than non-surrogate-reared pups and an age similar to their wild counterparts' (p.316) |

Asian elephants (*Elephas maximus*); bears (*Ursos arctos*); dingos (*Canis dingo*); octopods (*Amphioctopus marginatus*) and otters (*Enhydra lutris nereis*).

Of the 105 publications included in the database, 80.0% were on captive subjects, whereas 20.0% were wild observations. Fifteen different tool-types were found. Materials used as tools were primarily sticks: 60.0%, stones: 13.3% and a combination of sticks and stones: 1.9%. However, other objects were also used as tools: 3.8% boxes, 3.8% leaves, 2.9% cups, 2.9% water, 1.9% string, 1.9% doors, 1.9% hair, 1.9% paper, 1.0% coconuts, 1.0% hooks, 1.0% wires and 1.0% oyster shells.

Of all the animals included in the database, most reports were found on primate individual learning (73.3%; see above). Of the primates, most reports were from chimpanzee tool-use (35.1%), followed by gorillas (13.0%), orangutans (5.2%) and bonobos (2.6%). The remainder of the reports came from monkeys (see above; 44.2%). The second highest number of reports was found for bird species. Of these species, most reports concerned New Caledonian crows (33.3%), followed by rooks (19%) and Goffin cockatoos (14.3%).

With regards to the interpretation of the results, 50.4% of all the studies in Table 2 suggested that their species acquired the behaviour primarily via individual learning, 15.2% suggested social learning played a role (although the specific social learning mechanism is often not identified) and 34.2% made no comment on the learning mechanism at work. Within the total number of reports, 47.3% of studies on primates concluded that individual learning was driving the behaviour, 17.1% concluded that social learning played a major role (again, the specific social learning mechanism is rarely mentioned) and 35.5% did not mention the learning mechanism. In the studies on birds, 63.6% mention individual learning, 9.0% mentioned social learning and 27.2% made no comment. In studies with other animals, 42.8% suggested that individual learning was the main mechanism at play, 14.2% mentioned social learning and 42.8% made no comment.

## Findings of the literature review

The studies included in Table 2 were found following the approach described in the methods section. For most of the reports (50.4%) included in Table 2, the authors of each

paper described their observations primarily as the products of individual learning, therefore removing any subjective decisions made by the authors on the mechanisms behind the emergence of the behaviour (see column five of Table 2). Only 15.2% of the studies included in Table 2 favoured a social learning approach, and only five studies attempted to identify the type of social learning. Of these five studies, four suggested that stimulus enhancement (a type of non-copying social learning) played a role, and only one study—on Northern blue jays (*Cyanocitta cristata*; *Jones & Kamil, 1973*)—argued that the subjects were acquiring the behaviour through form copying (namely: imitation). The remaining papers in which the authors did not explicitly state mechanisms, were deemed to be best, and most parsimoniously, explained via individual learning for the first individuals who showed the behaviour (as after the behaviour emerged in a group, non-copying variants of social learning most likely influence the probability of acquisition of this behaviour in other group members; *Bandini & Tennie, 2017*, *2019*). For the cases in which the subjects were naïve to the behaviour before testing, received no demonstrations of the behaviour before or during testing, yet acquired the target behavioural form, the most likely explanation was deemed to be individual learning (however note that these cases were still marked as remaining neutral on the learning mechanisms underlying the behaviour, and constitute 34.2% of the studies included in Table 2). These findings suggest that the majority of studies on the emergence of a new tool-use behaviour across animal species favour an individual learning-driven explanation, with some further suggesting that non-copying social learning mechanisms (such as stimulus enhancement) facilitated the likelihood of acquisition of the behaviour, but only one article suggested that copying social learning played a role in the acquisition of the behaviour (see below for further discussion of this specific publication).

In the following section, case studies from most of the species included in Table 2 are discussed, alongside some of the implications of these reports. For the sake of brevity, not all the species or reports included in Table 2 are discussed in the following section. As most reports were found to be from primate and bird species (see also quantitative results section above), the following section focuses primarily on these species (however reports from other tool-using animals are also included).

## Individual learning in primates
### Chimpanzees
*Kummer & Goodall (1985)* describe one of the first instances of spontaneous tool-use in wild chimpanzees from Gombe, Tanzania. The new behaviour involved the use of sticks as levers to open banana boxes introduced by the researchers (*Kummer & Goodall, 1985*). Although the previous tool-use knowledge of the Gombe chimpanzees is not reported, this observation was included in Table 2 because the banana boxes were artificial objects introduced by the researchers, making it unlikely that the chimpanzees had already encountered this specific problem in the wild. Indeed, many of the first observations of tool-use in animals were the product of individuals interacting with novel, artificial or introduced materials and objects (*Köhler, 1925*; *Lefebvre, 1995*). Subsequently, many early researchers interpreted new observations of animal tool-use from an anthropocentric

point of view. Indeed, despite writing that the chimpanzees showed the behaviour 'independently' (i.e. without requiring social models), *Kummer & Goodall (1985)* also argue that the chimpanzees were learning the behavioural form from each other as several individuals demonstrated the same behaviour very quickly once they were exposed to the banana box. The authors interpret this observation as evidence that the chimpanzee must have copied the behaviour by watching others beforehand, rather than individually learning the behaviour[1]. Although it is possible that the chimpanzees observed each other, this fact alone might be rather irrelevant, as current experimental data suggests that unencultured chimpanzees do not spontaneously copy actions in most contexts (see above, but see also *Horner & Whiten (2005)* for an alternative view). Therefore, an alternative explanation, which is more consistent with the interpretations of the majority of the studies in Table 2 and the current experimental data, may be that the chimpanzees converged instead on the same behavioural form by individually developing the most efficient solution to the common problem at hand. This is not to say that other variants of social learning played no role—it is indeed likely that the chimpanzees were drawn to the banana box through stimulus and/or local enhancement when they observed other individuals interacting with the box.

However, it is only with more controlled cases of reinnovations that learning mechanisms can be identified with (more) confidence in wild populations. *Hobaiter et al. (2014)* report one of the rare occurrences in which the emergence of tool-use behaviours in wild chimpanzees was tracked as the frequencies increased within the community. The authors describe the emergence of two tool-use behaviours in the chimpanzees of Budongo Forest, Uganda: moss-sponging and leaf-sponge re-use. The tools were used to retrieve water from a waterhole that had recently been flooded (*Hobaiter et al., 2014*). The authors followed the increases in frequency of these two behaviours within the Sonso chimpanzee community. Using network-based diffusion analysis for the first behaviour, *Hobaiter et al. (2014)* attributed at least 85% of the newly observed events of moss-sponging to social learning, arguing that for each new observation, naïve chimpanzees enhanced their chances of developing moss-sponging by a factor of 15. Thus, non-copying social learning most likely played a role in increasing the frequency of this behaviour within the community. Yet, even though the frequency of the behaviour increased through non-copying social learning, this leaves open the question as to whether the form of the behaviour was individually or socially learnt. This study was included in the database because, after the original innovation of moss-sponging by the alpha male, the alpha female also developed the behaviour independently before having observed the male moss-sponging (researchers were able to closely observe the chimpanzees during the study period, allowing for conclusions to be made on the background knowledge of each of the group members; *Hobaiter et al., 2014*). Therefore, it seems possible that although non-copying social learning facilitates the individual acquisition of the behavioural form by the rest of the group, the form of moss-sponging can be individual learnt, as it was reported in two separate individuals in the absence of any social models. In a follow-up study on the spread of moss-sponging three years after its innovation, *Lamon et al. (2017)* agreed with the individual learning approach to explaining the increase in frequency of

[1] For example *Kummer & Goodall (1985*, 203*)* write: '*Primates are remarkably ill-equipped with innate technologies*'
this behaviour: '*Of course, each moss-sponger has to individually learn the behaviour, but in all likelihood, this was facilitated by the social influence exerted by other group members that acted as models*' (*Lamon et al., 2017*, 6).

On the other hand, for the second behaviour, leaf-sponge re-use, the social network analysis failed to show a role of social learning in the increase in frequency of the behaviour, as eight naïve chimpanzees independently acquired the same behavioural form (*Hobaiter et al., 2014*). Thus, this study suggests that both moss-sponging and leaf sponge-reuse seem to be within chimpanzees' individual learning abilities, but both behaviours are influenced to a higher or lesser extent by non-copying social learning.

Several similar reports of reinnovations of stick and leaf tool-use behaviours by naïve individuals were also found in captive chimpanzees. Wolfgang Köhler spent many years (1914–1920) investigating the cognition behind captive chimpanzee stick tool-use, and concluded that many of these behaviours, such as using a stick to retrieve out-of-reach foods, can emerge via '*insight learning*'[2]. Similarly, *Kitahara-Frisch & Norikoshi (1982)* examined the origins of sponge-making (a behaviour in which wild chimpanzees use leaves as sponges to absorb liquids such as water or honey; *Kitahara-Frisch & Norikoshi, 1982*, 42) and found that captive chimpanzees showed the same form of sponge-manufacture and use as their wild counterparts. The authors conclude that, contrary to previous claims, '*the example of the mother is by no means necessary for the habit to appear in young animals. This observation raises the question of whether the acquisition of so-called proto-cultural habits does not rely as much, at least, on independent reinvention as on transmission through imitation learning*'. The interpretation of the results of many of the studies included in Table 2 agree with those proposed by the ZLS[3] and provide mounting evidence for the view that simple stick tool-use is, most likely, within the individual learning capabilities of both wild and captive chimpanzees (*Bandini & Harrison, 2020*; see also *Gruber et al. (2009, 2011*) for reports of wild chimpanzees *not* acquiring stick tool-use, although see the section on behavioural flexibility below for further discussion of these studies).

### Gorillas

Although gorillas do not commonly use tools in the wild, sporadic cases of spontaneous tool-use by gorillas in the wild and in captivity have been reported. One such report comes from *Kinani & Zimmerman (2015)*, who describe, for the first time, a female juvenile gorilla (*Gorilla beringei beringei*) using a stick to fish for ants in Volcanoes National Park, Rwanda. The authors describe the behaviour as being similar to chimpanzee ant-dipping (which was classified as a '*putative cultural trait*'; *Whiten et al., 1999*; for chimpanzees before it was discovered that the environment plays an important role, alongside other factors, in shaping the form of the behaviour; *Humle & Matsuzawa, 2002*, *Humle, 2006*, *Schöning et al., 2008*). This is the first report of ant-dipping (and indeed, any stick tool-use) in wild gorillas. Further evidence of gorillas' spontaneous tool-use is offered by *Lonsdorf et al. (2009)*, who exposed captive gorillas to an artificial termite mound in their enclosure. The authors report that although not all the gorillas in the group used tools, the alpha male fished for the bait using a stick on the first day of the study, demonstrating

[2] We would now include 'insight learning' within 'individual learning', as the term 'insight' could be taken to imply a sole, or primary, role of genetics in the emergence of a behaviour. However, current evidence now suggests that genetics play a role alongside other factors such as the environment and individual and social learning (*Reindl, Bandini & Tennie, 2018*).

[3] For example see *Anderson (1985)*, *Auersperg et al. (2012)*, *Auersperg et al. (2011)*, *Bandini & Tennie (2019, 2017)*, *Beck (1966)*, *Breuer, Ndoundou-Hockemba & Fishlock (2005)*, *Epstein (1985)*, *Foerder et al. (2011)*, *Fontaine, Moisson & Wickings (1995)*, *Kenward et al. (2005)*, *Kitahara-Frisch & Norikoshi (1982)*, *Köhler (1925)*, *Laumer et al. (2017)*, *Mendes, Hanus & Call (2007)*, *Morgan & Abwe (2006)*, *Morimura (2003)*, *Neadle, Allritz & Tennie (2017)*, *Pouydebat et al. (2005)*, *Rutz et al. (2016)*, *Tokida et al. (1994)*, *Visalberghi, Fragaszy & Savage-Rumbaugh (1995)*, *Visalberghi & Trinca (1989)*, *Weir (2002)*, *Westergaard & Suomi (1994b)*, *Yamamoto et al. (2008)*.

the same behavioural form as wild chimpanzees, despite being naïve (*Lonsdorf et al., 2009*). Similarly, *Nakamichi (1999)* observed a western lowland gorilla (*Gorilla gorilla gorilla*) at the San Diego Wild Animal Park throw sticks into the foliage of trees to knock down leaves and seeds (which were later consumed), and *Fontaine, Moisson & Wickings (1995)* describe how a group of gorillas in a zoo in Gabon spontaneously used sticks to reach objects outside of their enclosure and coconut fibres as sponges to absorb water, demonstrating similar behavioural forms as chimpanzees and other primates who regularly use tools (*Fontaine, Moisson & Wickings, 1995*). Although the behaviours described above are not particularly complex (following the current definition in the literature; *Meulman et al., 2012*), these reports demonstrate that gorillas are capable of spontaneously using some tools when motivated. Thus, despite rarely showing tool-use behaviours in the wild, gorillas have demonstrated a surprisingly extensive ability to innovate and reinnovate various tool-use behaviours via individual learning (*Boysen et al., 1999*; *Fontaine, Moisson & Wickings, 1995*; *Lonsdorf et al., 2009*; *Manrique, Völter & Call, 2013*; *Nakamichi, 1999*; *Natale, Poti' & Spinozzi, 1988*; *Neadle, Allritz & Tennie, 2017*; *Pouydebat et al., 2005*; *Tennie et al., 2008*).

### Capuchins

Capuchins exhibit one of the most extensive natural tool-use repertoires in primates (second only to chimpanzees), and a clear ability to individually learn these repertoires, including stone-tool behaviours. One example of stone-tool behaviour comes from wild bearded capuchins (*Sapajus libidinosus*) in Serra da Capivara National Park in Brazil. These capuchins have been observed deliberately pounding standing conglomerates with smaller hammerstones to break open fragments of the larger stones (*Proffitt et al., 2016*). This behaviour, named stone-on-stone (SoS) percussion, involves an individual selecting a smaller pounding stone and using it to strike the cobbles embedded in a conglomerate of standing stones using one or both hands (*Proffitt et al., 2016*).
The purpose behind SoS percussion is still unclear, but it may be that the monkeys break stones open to ingest powdered lichens or quartz from inside (support for this possibility comes from a report of a captive capuchin (*Cebus nigritus*) that was also observed practicing SoS and then licking the lichens inside the stone; *Bortolini & Bicca-Marques, 2007*). An interesting by-product of this behaviour is that by pounding the stones together, the capuchins produce flakes superficially similar to those made by early hominins (according to the authors; *Proffitt et al., 2016*). This is the first observation of SoS behaviour, and the production of flakes, in wild capuchins. Although the learning mechanisms behind the acquisition of this behaviour in the wild remain to be identified, an earlier experimental test with captive capuchins (*Sapajus apella*) describes how a similar stone pounding behaviour, including the production (and use) of flakes, was reinnovated spontaneously by naïve capuchins (*Westergaard & Suomi, 1994a*).
The unenculturated captive capuchins tested by *Westergaard & Suomi (1994a*; the unenculturated status of the capuchins was confirmed by G. Westergaard, 2019, personal communication*)* were also able to use the flakes they made as tools, and used them to cut open (by pushing and chiselling) the acetate top of a baited testing apparatus

(similarly to the testing conditions the enculturated bonobo Kanzi faced; *Toth et al., 1993*). This study is particularly interesting as it demonstrates that naïve capuchins are not only able to reinnovate a wild behaviour (SoS percussion), but can also use the flakes they produced.

*Ottoni & Mannu (2001)* also describe a stone-tool behaviour in wild capuchins. The wild bearded capuchins of the Ecological Park of the Tiete River (São Paulo, Brazil) are provisioned daily with fruit and protein, but spend the majority of the day foraging for naturally occurring food sources (*Ottoni & Mannu, 2001*). During one of these foraging sessions, the monkeys were observed cracking open mature *Syagrus* nuts using a hammerstone on an anvil, similarly to how wild chimpanzees crack nuts. The authors conclude that nut-cracking is driven by '*some kind of observational learning, albeit restricted to stimulus enhancement, as a starting point of a long process of individual improvement by trial-and-error*' (*Ottoni & Mannu, 2001*, 357). Similarly, in a study on nut-cracking in captive capuchins, *Visalberghi (1987)* provided naïve subjects with stones and encased almonds, but no social information on the behaviour. Over a period of ten trials, two males successfully used stone tools to crack open the almonds (*Visalberghi, 1987*). Although the rest of the group (40 individuals) had ample opportunities to observe the two males cracking nuts, no other individual from the group reinnovated nut-cracking (see below for further discussion). Nut-cracking has now also been observed in other species of capuchins, including wild *Cebus libidinosus* (*Waga et al., 2006*), *Sapajus apella* (*Izawa & Mizuno, 1977*), *Cebus apella* (*Struhsaker & Leland, 1977*), and in another captive population of *Cebus apella* in captivity (*Antinucci & Visalberghi, 1986*). These findings strongly support the view that nut-cracking is within the individual learning abilities of several capuchin species. Various other spontaneous tool-use behaviours have been observed in capuchins (*Boinski, 1988*; *Fernandes, 1991*; *Soares Bortolini & Bicca-Marques, 2007*; *Visalberghi & Trinca, 1989*; *Westergaard & Fragaszy, 1987*), making capuchins a promising species for the study of individual learning in primate tool-use.

### Macaques

Some populations of *Macaca fascicularis aurea* (*Mfa*), a subspecies of long-tailed macaques, frequently use stone tools and two different methods (pound-hammering and axe-hammering) to pound open encased foods in Southeast Asia (*Malaivijitnond & Hamada, 2008*). The closely related subspecies, *Macaca fascicularis fascicularis* (*Mff*), have, however, never been observed to use tools, despite sharing an environment and even interbreeding with *Mfa* (*Bandini & Tennie, 2018*; *Falótico et al., 2017*; *Gumert, Kluck & Malaivijitnond, 2009*). Despite having access to social information on tool-use (i.e. *Mfa* who could act as demonstrators of the tool-use behaviours), *Mff* continue not to use stone tools in the wild. However, *Zuberbühler et al. (1996)* observed a captive *Mff* spontaneously using a stick to retrieve apples that had fallen just out of reach outside the enclosure. The behaviour was then also observed in other members of the group. Stick manipulation was found to slightly increase when the original innovator was raking in

apples. However, the small increase of manipulation of sticks (perhaps via social facilitation) did not always result in tool-use by the other members of the group. The authors therefore conclude: '*We cannot conclude from these data that stimulus enhancement is a necessary prerequisite to becoming a skilled animal, because MD, the inventor of the technique, most likely developed his skill by means of individual learning*' (*Zuberbühler et al., 1996*, 10). Naïve *Mff* have also been observed using human hair as dental floss (*Watanabe, Urasopon & Malaivijitnond, 2007*), and Tonkean macaques (*Macaca tonkeana*) have been observed spontaneously using sticks to retrieve honey from an out-of-reach apparatus (*Anderson, 1985*), to rake food into the enclosure (*Ueno & Fujita, 1998*), and making climbing structures out of sticks and browse (*Ducoing & Thierry, 2005*; *Westergaard & Lindquist, 1987*). Japanese macaques (*Macaca fuscata*) also use human hair as dental floss (*Leca, Gunst & Huffman, 2010*) and use stones (and infants) to push fruit out of a tube (*Tokida et al., 1994*). Lion-tailed macaques (*Macaca silenus*) use probes to extract syrup from a baited apparatus (*Westergaard, 1988*) and Rhesus macaques (*Macaca mulatta*) use cup-like containers to transport water around their enclosures (*Parks & Novak, 1993*). These reports demonstrate that, similarly to gorillas (who do not often practice tool-use in the wild), various species of macaques can reinnovate some tool-use behaviours when in the appropriate context.

## Individual learning in birds

### Crows

In comparison to primates, the case for individual learning in bird material culture seems to be somewhat less debated. Indeed, several accounts exist of naïve birds in the wild and in captivity spontaneously acquiring wild behaviours (*Auersperg et al., 2012*; *Bird & Emery, 2009a*; *Collias & Collias, 1964*; *Epstein, 1985*; *Jones & Kamil, 1973*; *Overington et al., 2011*; *Taylor et al., 2010*; *Tebbich et al., 2001*, *2007*). Some of the most impressive accounts of the individual learning of tool-use come from New Caledonian crows (*Corvus moneduloides)*, who possess sophisticated stick tool-use repertoires, rivalling even those of non-human primates (*Weir, 2002*). Similarly to chimpanzees, some New Caledonian crow tool-use is subject to regional variation. In the chimpanzee case however, this variation has been used as evidence for the view that the behavioural forms are dependent on social learning (*Whiten et al., 1999*, *2001*; *Gruber et al., 2015*). Yet this claim has not been made as often for wild New Caledonian crows (*Weir & Kacelnik, 2006*; although see also *Logan et al. (2016)* for an experimental test on the non-copying social learning abilities of New Caledonian crows and *Hunt & Gray (2003)* for claims that New Caledonian crows may have cumulative culture). Experimental studies with captive New Caledonian crows have demonstrated that naïve crows can spontaneously make and use some of the same tools as their wild counterparts. The authors state: '*In the light of our findings, it is possible that the high level of skill observed in wild adult crows is not socially acquired*' (*Kenward et al., 2005*, 121). Naïve New Caledonian crows are also capable of developing tool bending (*Weir & Kacelnik, 2006*) and metatool-use (i.e. the ability to use one tool on another; *Taylor et al., 2007*; *Whiten & Byrne, 1997*). These findings led

*Kenward et al. (2005*, 121*)* to conclude that: '*the ability of this species to manufacture and use tools is at least partly inherited and not dependent on social input*'.

Another strong example of individual learning in birds comes from Hawaiian crows (Alalā; *C. hawaiiensis)*. Hawaiian crows are extinct in the wild and currently exist only in captivity, but it is likely that in their past natural state these crows also used tools (similarly to New Caledonian crows; *Rutz et al., 2016*). In an experimental study with captive Hawaiian crows, 78% of the population spontaneously used tools to probe for out-of-reach food, without social demonstrations (*Rutz et al., 2016*). Similarly to *Kenward et al. (2005)* conclusion for New Caledonian crows, *Rutz et al. (2016*, 405*)* also suggest that the observed behavioural repertoire is therefore a product of individual learning and/or genetic predispositions: '*Alalā clearly possess a propensity to "discover" tool-assisted foraging solutions independently*'.

### Other birds

Similarly to primates, several species of birds that do not use tools in the wild spontaneously acquire tool-use when provided with the materials in captivity. One example of this phenomenon are captive born and raised blue jays (*Cyanocitta cristata*), who do not use tools in the wild, but were observed tearing up pieces of newspaper and using them to rake out-of-reach food pellets from outside their cage (*Jones & Kamil, 1973*). Although note that this is the only study in Table 2 in which the authors interpreted their observation as the product of imitation ('*the fact that to date we have found six jays in our colony demonstrating tool-using behaviour is more likely the result of […] observational learning or imitation than the result of the independent acquisition of this behaviour by each of the six jays*', *Jones & Kamil, 1973*, 1078). In the light of new understanding on the role of individual learning and non-copying social learning in bird tool-use, it is also possible that mechanisms other than copying drove the emergence of this behaviour, but this remains to be tested.

Most other cases of spontaneous tool-use in birds have been cited as examples of individual learning rather than of imitation and/or other variants of copying social learning. For example when captive Goffins cockatoos were observed making and using stick tools to rake in food from outside the enclosure, the authors argue that their '*observations prove that innovative tool-related problem-solving is within this species' cognitive resources*' (*Auersperg et al., 2012*, 2). When captive Rooks (*Corvus frugilegus*) used stones to collapse a platform to retrieve a worm, the authors similarly concluded that the reinnovation was an individually learnt solution to a new problem (*Bird & Emery, 2009b*). Similarly, when naïve pigeons solved a task inspired by *Köhler (1925)* work with chimpanzees, in which the pigeons had to use boxes to reach a banana hanging outside their enclosure, the author concludes that they '*have on hand an instance of insightful problem solving*' (*Epstein, 1985*, 62). Indeed, albeit some minor exceptions, contrary to the primate case, the view that bird tool-use is the product of individual learning is pervasive throughout the literature (*Laumer et al., 2017*; *Tebbich & Bshary, 2004*; *Tebbich et al., 2001*, *2007*).

### Individual learning in other animals

Although most reports of animal spontaneous tool-use come from primates and birds, other animal species have also demonstrated the ability to individually acquire tool-use behaviours. For example a dingo (*Canis dingo*) was recorded moving objects around his enclosure, including a table, to climb on to reach food or to observe other animals outside his enclosure (*Smith, Appleby & Litchfield, 2012*). The dingo's behaviour (using a table to access out-of-reach food) is reminiscent of *Köhler (1925)* early studies in which chimpanzees used boxes to reach hanging bananas. Similarly, reports exists of both wild and captive bears using tools, including one experimental study which examined whether captive bears would manipulate and re-orient objects to climb on to reach suspended food (*Deecke, 2012*; *Waroff et al., 2017*). Furthermore, *Chevalier-Skolnikoff & Liska (1993)* describe over 20 tool-use behaviours that emerged without social learning by captive African (*Loxodonta africana*) and Asian elephants (*Elephas maximus*). Behaviours included: '*reach toward food with stick held in the trunk*'; '*Rub the body with a stick*'; '*probe musth gland with stick*' (*Chevalier-Skolnikoff & Liska, 1993*, 210). A recent observational report describes the use of two coconut halves as a protective shell by an octopus (*Amphioctopus marginatus*; *Finn, Tregenza & Norman, 2009*), and orphaned, captive juvenile sea otters (*Enhydra lutris nereis*) were found to develop the same stone tool pounding behavioural forms as observed in wild adult otters (*Nicholson et al., 2007*). Collectively, these studies demonstrate that tool-use is not restricted to primates and birds, but that many other animals possess the ability to use tools, and crucially, the ability to individually learn their behavioural forms.

## DISCUSSION

The studies complied and discussed in this review emphasise the extent of individual learning for the acquisition of tool-use behaviours across various species. However, the studies included in Table 2 also demonstrate that not all individuals, despite being capable of doing so, will acquire all the behaviours within their potential tool-use repertoires. Some of the factors that also influence whether naïve subjects develop tool-use are discussed below.

### Social learning

Although the studies described in this review highlight the importance of individual learning in the acquisition of tool-use behavioural forms, this review does *not* suggest that social learning plays no role in the development and maintenance of animal tool-use repertoires. On the contrary, many of the studies in Table 2 that included social learning conditions found evidence of non-copying social learning mechanisms regulating the frequency of behaviours, and speed of acquisition, once one, or more, individuals in a group had developed the target behavioural form (*Horner & Whiten, 2005*; *Hopper et al., 2007*). Furthermore, just by including new apparatuses or objects into naïve subjects' enclosures, non-copying social learning (such as stimulus enhancement) is already involved in encouraging the subjects to explore and manipulate the new objects (*Bandini et al., 2020b*). Therefore, for these types of experimental studies, non-copying social learning

is often impossible to exclude from the testing conditions. It is therefore incontestable that these variants of social learning greatly facilitate the frequency of emergence of behaviours. However, it is also important to note that these mechanisms are not infallible, and despite opportunities for both individual and social learning, some group members may never express all the target behaviours within their repertoires. In fact, some of the studies in Table 2 describe the emergence of a behaviour in only some of the subjects within their samples, whilst other members of the group never express the behaviour despite ample exposure to the behavioural models. Similarly, evidence from wild chimpanzees also suggests that few innovations 'catch-on', despite opportunities for social learning (*Nishida, Matsusaka & McGrew, 2009*; *Bandini & Harrison, 2020*). *Tebbich et al. (2001)* directly examined this phenomenon by measuring the rates of acquisition of a novel tool-use behaviour in naïve finches when exposed to a model and in a control group. The authors conclude: '*the presence of a model does not influence the ontogeny of tool-use: this behaviour was expressed in the absence of a model and the development was not slower without than with a model*' (*Tebbich et al., 2001*, 2192). Several other studies report similar results across various species[4]. Collectively, these findings suggest that alongside individual learning and non-copying social learning, other factors must also play an important role in the emergence of tool-use behaviours (*Bandini & Tennie, 2018*).

## Environment

The environment is one of the most important contributing factors to the reinnovation of behavioural forms. Species with the most extensive tool-use repertoires are often ones that have invaded new niches (*Alcock, 1972*). Colonising new areas requires the innovation of behaviours and foraging techniques, placing the species under new selective pressures (*Miller, 1956*). These acquired characteristics then can even shape the environment itself (e.g. 'niche-construction' theory; *Odling-Smee, Laland & Feldman, 1996*), resulting in the environment playing a role as both an explanation and explanandum for animal tool-use. One concrete example of how the environment can influence the likelihood of tool-use emerging is encapsulated by the concept of the 'captivity effect' (*van Schaik, Deaner & Merrill, 1999*). The captivity effect describes the observation that some captive animals seem to outperform their wild counterparts in both the diversity and frequency of behaviours, such as tool-use (*van Schaik, Deaner & Merrill, 1999*). The safer and more predictable environment provided by captivity, regular provisioning, the lack of predators, and higher levels of free time and energy afforded by captive settings have all been cited as factors behind the increased levels of tool-use in captive animals (*Haslam, 2013*). Thus, among other factors, the increased opportunities to explore new behaviours afforded by captive environments most likely enhance the likelihood that animals will develop behaviours within their repertoires. However, it seems unlikely that the captivity effect enhances animals' cognition to such a high level to allow them to acquire behavioural forms outside of the capability of their wild counterparts (i.e. outside of their species' ZLS; *Reindl, Bandini & Tennie, 2018*). Indeed, only highly enculturated subjects seem capable of going beyond their species' 'natural' capacities (*Reindl, Bandini & Tennie, 2018*). Therefore, animals living in conspecific group settings in

---

[4] For example: *Anderson (1985)*, *Antinucci & Visalberghi (1986)*, *Bandini & Tennie (2018)*, *Beck (1966)*, *Biro, Haslam & Rutz (2013)*, *Geissmann (2009)*, *Hayashi, Mizuno & Matsuzawa (2005)*, *Hirata, Morimura & Houki (2009)*, *Menzel, Davenport & Rogers (1970)*, *Nakamichi (1999)*, *Overington et al. (2011)*, *Smith, Appleby & Litchfield (2012)*, *Sumita, Kitahara-Frisch & Norikoshi (1985)*, *Tebbich et al. (2001)*, *Tokida et al. (1994)*, *Vernes et al. (2007)*, *Visalberghi, Fragaszy & Savage-Rumbaugh (1995)*, *Visalberghi & Trinca (1989)*, *Yamamoto et al. (2008)*, *Zuberbühler et al. (1996)*.

captivity, with limited human contact, seem merely more likely to acquire tool-use behaviours than their wild counterparts, but these behaviours would, hypothetically, be within the reach of all individuals of the same species (*Bandini & Tennie, 2018*).

With regards to the effect of the environment in the wild, differences may also influence the emergence of tool-use behavioural forms. To explain the advent of tool-use in wild species, some have argued that encounter rates with certain food types or resources greatly encourage the emergence of associated tool-use behaviours in wild animals (the *opportunity hypothesis*; *Koops, McGrew & Matsuzawa, 2013*; *Koops, Visalberghi & Van Schaik, 2014*). Others have argued instead that tool-use emerges as a direct response to times of scarcity of preferred food sources (the *necessity hypothesis*; *Fox et al., 2004*; *Koops, Visalberghi & Van Schaik, 2014*). *Koops, McGrew & Matsuzawa (2013)* and *Koops, Visalberghi & Van Schaik (2014)* directly tested the two opposing hypotheses on wild primates (focusing especially on chimpanzees) and found that access to the appropriate resources and food sources (i.e. the opportunity hypothesis) was the most compelling explanation for the regional variation observed in chimpanzee tool-use behaviours. The opportunity hypothesis may therefore explain at least parts of both the regional variation in tool-use repertoires observed in some species of wild primates and the increased levels of tool-use behaviours observed in captive animals.

## Genetic influences

The role of genetics in animal tool-use is still relatively poorly understood. *Langergraber et al. (2011)* indirectly examined the role of genetic influence on the 39 chimpanzee tool-use behaviours identified by *Whiten et al. (1999*, *2001)* as 'cultural' (i.e. relying on social learning, according to *Whiten et al., 1999*, *2001*) and concluded that '*genetic differences cannot be excluded as playing a major role in structuring patterns of behavioural variation among chimpanzee groups*' (*Langergraber et al., 2011*, 409). However, using cladistic analysis on the same 39 chimpanzee behaviours, *Lycett, Collard & McGrew (2007*, 547*)*, argue instead that their data '*support the suggestion that the behavioural patterns are the product of social learning and, therefore, can be considered cultural*'. These two contrasting studies demonstrate that the role of genetics in animal tool-use is still unclear and heavily debated. Yet, it is likely that alongside the environment, genetics play an important role in the emergence of behaviours. This effect is most clearly observed in studies in which very closely related subspecies were found to differ in their tool-use abilities (e.g. long-tailed macaques; *Luncz et al., 2017*; and otters; *Ladds, Hoppitt & Boogert, 2017*; *Bandini et al., 2020a*) even when they share the same environment and/or are placed in the same testing conditions. Furthermore, measurements of intelligence (which may be correlated to the ability to use tools; *Navarrete et al., 2016*) have also been suggested to be heritable in chimpanzees (*Hopkins, Russell & Schaeffer, 2014*), and dolphin sponging behaviour (in which dolphins use sponges as foraging tools; *Krützen et al., 2005*), has been suggested to be predicted by genetic relatedness with other spongers in the community (*Krützen et al., 2005*; although see *Sargeant et al. (2007)* for a contrasting view). Furthermore, the line between genetic predispositions for behaviours and individual learning of these forms is often blurry, and sometimes impossible to define. Indeed, although

it is unlikely that many behavioural forms are genetically 'imprinted' in animals, it is very possible that certain species are genetically predisposed to pay attention to specific types of stimuli, which then necessarily, or very likely, lead to a tool-use behaviour emerging (i.e. these behavioural forms would then be exaptations, rather than adaptations; *Bandini et al., 2020b*). In summary, further research is still required into the role of genetic differences in the tool-use abilities of animals. In the meanwhile, this factor should be kept in mind as an influence on the acquisition of tool-use in animals, even on a subspecies level.

## Pre-existing techniques

An individual's background knowledge of the materials of the target behavioural form may also influence the acquisition of that behaviour in naïve animals. *Gruber et al. (2009*, 2011*)* work with the Sonso and Kanywara chimpanzee communities in Uganda provides relevant data for discussions on the role of these factors in the acquisition of a novel behaviour. Despite neighbouring each other, the two communities adopt different methods to acquire honey from trees: Sonso chimpanzees most often use their hands or leaves to access the honey, whilst Kanywara chimpanzees generally use stick tools (*Gruber et al., 2009*). To examine the stability of these differences, *Gruber et al. (2009)* placed an artificial log with two honey-filled cavities in both groups, to encourage the Sonso chimpanzees to switch their method to the Kanywara stick-tool-use approach. However, contrary to expectations, the Sonso chimpanzees remained with their pre-existing technique of using leaves to scoop up the fluid, even from the narrower cavities of the artificial log, and the Kanywara chimpanzees continued to use sticks for the remainder of the experiment. The stability in methods remained even after the researchers placed sticks inside the holes in the Sonso chimpanzees' log (the chimpanzees simply removed the sticks and continued to use leaves to absorb the honey). *Gruber et al. (2009*, 1809*)* suggest that the Sonso chimpanzees' reliance on community-specific techniques, rather than switching to stick tool-use '*supports a culturally based rather than an individual acquisition of the behaviour*'. The authors further argue that the Sonso chimpanzees are '*unable*' to use sticks because they have never observed another individual using a stick to retrieve honey. On the other hand, the Kanywara chimpanzees, who regularly use sticks, have ample opportunities for naïve individuals to socially learn stick tool-use for honey dipping (*Gruber et al., 2009*). Whilst this may indeed be the case, an alternative explanation for these findings is that, whilst non-copying social learning may have encouraged the chimpanzees to initially adopt either a leaf or stick based approach to this task (simple increases in exposure to either leaves or sticks around the problem-space most likely influences which method an individual chooses; e.g. 'cultural founder effects'; *Tennie, Call & Tomasello, 2009*; *Tennie et al., 2011*), both behaviours are within the chimpanzees' individual learning abilities (and therefore all individuals, in both communities, are technically capable of developing either technique). Thus, although capable of using sticks, the Sonso chimpanzees may have not switched to stick-use simply because they already had a pre-existing, efficient, technique to reach the same end goal (*Tennie et al., 2011*). Indeed, chimpanzees often seem hesitant to switch to a new technique if it is not considerably more efficient than their previous method (*Davis et al., 2016*; *Harrison &*

*Whiten, 2018*; *Hrubesch, Preuschoft & Van Schaik, 2009*; *Bandini & Harrison, 2020*). The existence of an already efficient technique may hinder the exploration of new methods if the end result is the same, and the new method is not vastly more efficient (no data exists on differences in efficiency between the leaf and stick methods, but it is likely that if differences do exist, they are minimal and thus hard to observe for this behaviour; *Tennie et al., 2011*). Therefore, the chimpanzees' relative inflexibility observed in *Gruber et al. (2009)* work may not only be a reflection of the Sonso chimpanzees' dependence on social learning, but rather may reveal a form of 'functional fixedness', in which the subjects' past experience with objects in different contexts hinder their ability to develop alternative behavioural forms with these tools (functional fixedness via individual learning; *Hanus et al., 2011*). This phenomenon may, therefore, have obstructed the Sonso chimpanzees from switching to a new method with tools that they already used for other purposes. This inflexibility may constitute a limiting factor for non-human tool-use (*Brosnan & Hopper, 2014*).

Previous experience with the materials of a behaviour does not, however, always play a limiting role on the reinnovation of behaviours. Indeed, several studies have demonstrated that the opportunity to manipulate components of a behaviour for an extended period (or during an ontogenetic sensitive learning period) is beneficial, and sometimes even a necessity, for subsequent emergence of the behavioural form. For example juvenile chimpanzees (*Pan troglodytes*), macaques (*Macaca fascicularis*), and sea otters (*Enhydra lutis*) who were observed to manipulate and handle stones from a young age, were all found to be more likely to develop stone tool-use behaviours later on in life (*Birch, 1945*; *Biro et al., 2003*; *Tan, 2017*; *Tebbich et al., 2001*). Furthermore, some innovations across species may be the result of generalisations from previous experience with similar tools or problems, and therefore having pre-existing experience with a tool may increase and individual's likelihood of developing a seemingly new behavioural form (see also *Tennie, Hopper & Van Schaik, 2020*).

## Other cognitive mechanisms

Although the focus of this review has been on individual and social learning, other cognitive mechanisms also play important roles in the acquisition of tool-use behaviours. Working memory, motivation and attention to novel objects and object manipulation, physical cognition, causal reasoning and information processing have all been cited in the literature as essential cognitive abilities for tool-use (see also *Sanz, Call & Boesch, 2013*; *Seed & Byrne, 2010*, *Read, 2008*). *Sanz, Call & Boesch (2013)* identifies three 'key ingredients' for animal tool-use: (1) information seeking and hoarding, (2) object manipulation and (3) problem solving abilities. Alongside these abilities, working memory has been cited as a key requirement for animal tool-use, and perhaps a limiting factor for some species (*Read, 2008*). Therefore, alongside the ability to learn individually and socially, a suite of cognitive abilities may be required for animals to drive and sustain their tool-use repertoires.
## CONCLUSIONS

This review has provided evidence from both wild and captive animals to support the view that many tool-use behaviours can be learnt individually, and that non-copying social learning facilitates the frequency and stability of these behaviours within and across populations. The ZLS hypothesis is currently the most prominent in explicitly describing this approach for animal tool-use; however, several studies (as discussed above and in Table 2) have already offered data and interpretations consistent with this view. Indeed, several early animal behaviour researchers concluded from their own work that individual learning plays a pivotal role in tool-use: '*Even the highest vertebrates, primates, have certain […] forms of activity available, without specific training to develop them. They are present uniformly in all individuals of the same age group, and are invariably displayed if the general condition of the animals favours them*' (*Menzel, Davenport & Rogers, 1970*, 281).

Although individual learning may play an important role in encouraging the emergence of novel behaviours, it is the combination of individual learning and the various types of non-copying social learning that allow for the successful sustenance of the rich animal tool-use behavioural repertoires (*Reader & Laland, 2003*; *Barrett et al., 2018*; *Heyes, 2011*). Furthermore, other factors, such as the environment, genetics, pre-existing techniques and other cognitive abilities all contribute to the likelihood of a behaviour being reinnovated. The importance of these factors, alongside recognising the equifinality of behaviours (i.e. behaviours can emerge via different mechanisms, or a combination of various mechanisms, across individuals; *Barrett et al., 2018*), should not be neglected. The findings of this review suggest that whilst individual learning and non-copying social learning mechanisms are widespread across animal species, and indeed we should perhaps assume *a priori* that most animals do acquire some of their information via non-copying social learning, the evidence for form copying social learning in most animal species is much less clear. Therefore, future studies should continue to explicitly test for the role of copying in the acquisition of behavioural (and artefact) forms, ensuring that the experimental paradigms used are appropriate and can effectively identify specific copying mechanisms, rather than generic social learning (*Tennie, Call & Tomasello, 2012*; *Tennie et al., 2017*; *Bandini et al., 2020b*). So far, great apes have been the targets of most of these tests, as perhaps due to their close phylogenetic ties to humans (but also their reputation), they are often hypothesised to be the most likely candidates for having copying abilities. So far, however, evidence from great apes suggests that they rarely engage in form copying, and only really do so when trained extensively and/or when enculturated (*Tennie, Call & Tomasello, 2012*; *Tennie et al., 2017*; *Pope et al., 2017*). Future studies should continue to test unenculturated, untrained great apes, but also other animal species that may, ultimately, be found to possess copying abilities. Indeed, animals that perform songs may be more likely candidate species for acquiring and transmitting behavioural forms through copying social learning (e.g. whales, *Garland et al., 2011*). In the meantime, we strongly suggest to include individual learning, often cued via non-copying social learning, in the list of important drivers of animal tool-use behavioural forms .

As *Byrne (2007*, 285*)* elegantly asks: '*If a habit can be invented multiple times, perhaps it can be invented by every individual that has a real need of it?*'. This review has focused only on examining the role of individual learning of the tool-use repertoires of animals. However, reinnovations can occur across domains (e.g. gestures, vocalisations and social behaviours). Although the emphasis on copying is most evident for tool-use behavioural forms, animal behaviour researchers should examine the role of individual and social learning in other domains, as it is possible that individual learning may be an important driver in these other contexts as well (e.g. see A. Motes-Rodrigo et al., 2020, unpublished data).

## ACKNOWLEDGEMENTS

The authors are grateful to Alba Motes-Rodrigo and Damien Neadle for comments on an earlier draft of this manuscript. The authors are also grateful to the handling editor at PeerJ and the two anonymous reviewers for their constructive comments and suggestions.

### Funding

The authors are supported by the Institutional Strategy of the University of Tübingen (Deutsche Forschungsgemeinschaft, ZUK 63). At the time of writing, Claudio Tennie was supported by the ERC. This project has received funding from the European Research Council (ERC) under the European Union's Horizon 2020 research and innovation programme (grant agreement no 714658; STONECULT project). The funders had no role in study design, data collection and analysis, decision to publish, or preparation of the manuscript.

### Grant Disclosures

The following grant information was disclosed by the authors:
Institutional Strategy of the University of Tübingen (Deutsche Forschungsgemeinschaft, ZUK 63).
European Research Council (ERC): no 714658.

### Competing Interests

The authors declare that they have no competing interests.

### Author Contributions

- Elisa Bandini conceived and designed the experiments, performed the experiments, analysed the data, prepared figures and/or tables, authored or reviewed drafts of the paper, and approved the final draft.
- Claudio Tennie conceived and designed the experiments, authored or reviewed drafts of the paper, and approved the final draft.

### Data Availability

 There is no raw data or code associated with this review.

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
