# Peer review of "Exploring the role of individual learning in animal tool-use"

_PeerJ, doi:10.7717/peerj.9877_

## Round 0.1 · original submission · Minor Revisions

I have been extremely fortunate to receive two very thorough and helpful reviews from experts on this topic, both of whom had overall favorable impressions of your review. Therefore, I will not add substantive comments of my own but will ask you to carefully address the comments raised by the reviewers in a revision. I agree with Reviewer 1 that it is essential for you to establish definitions and criteria for your concepts that will be clear to naive readers.

I do have a very few minor comments:

On line 138, I have to be skeptical that one can know what kind of experience wild animals might have had with tools or similar objects prior to the observation.

You seem to have undertaken a very thorough and unbiased literature search but you do not mention the work on tool use in bears, of which there are at least three reports. This seems like an unfortunate omission as they represent a previously neglected group that may help generalize the findings further to less social species.

A lot of the discussion of previous findings relies on speculation as to how behaviors were learned. Please be more cautious in attributing individual learning if social learning cannot be confidently ruled out. Also, please keep in mind that, even in the absence of experience with exact tools, a certain degree of generalization can be expected from similar objects and contexts such that what appears to be novel may be simply generalization.

Do not use '&' within the main text, outside of parentheses.
and one slightly more significant comment:

I am struggling with the theoretical importance of distinguishing between individual learning and social learning as mechanisms underlying tool use. This should be made clear at the outset and in the conclusions. The paper would make a stronger impact if the goal went beyond outlining cases of individual learning. If you could say more about how this would impact our view of previous work that was considered as evidence of social learning or predictions regarding social versus asocial species so that the paper was hypothesis-driven rather than descriptive, it would give it more weight.

Reviewer 1 ·

Basic reporting

1. Is the review of broad and cross-disciplinary interest and within the scope of the journal?
Yes, the review is relevant to researchers from a variety of disciplines studying social and individual learning in animals, the origins of (non-) human culture, cultural evolution, and innovation. The review is within the scope of PeerJ, as its topic relates to a number of previously published papers in PeerJ.

2. Has the field been reviewed recently? If so, is there a good reason for this review (different point of view, accessible to a different audience, etc.)?
To my knowledge, a review bringing together data on individually learned tool-use behaviours in animals has not been done yet. There is a slight overlap to a recently published review on innovation in chimpanzees (Bandini & Harrison, 2020), but the two reviews have clearly different foci and unique contributions to the literature.

3. Does the Introduction adequately introduce the subject and make it clear who the audience is/what the motivation is?
The introduction is well-written, provides sufficient background information on the current state of the field and clearly describes the motivation for the current review.

Experimental design

1. Is the Survey Methodology consistent with a comprehensive, unbiased coverage of the subject? If not, what is missing?
Yes, the authors made sure to carry out the literature search in an unbiased manner.

2. Are sources adequately cited? Quoted or paraphrased as appropriate?
Sources are adequately cited/quoted.

Is the review organized logically into coherent paragraphs/subsections?
Yes, the review is organized logically. However, it would be helpful if the sources in Table 1 were sorted alphabetically within each species.

Validity of the findings

1. Is there a well developed and supported argument that meets the goals set out in the Introduction?
The review meets the goal set out in the introduction (to provide a dataset with cases of individually learned tool-use behaviours) and shows that, as argued in the introduction, the role of individual reinnovations for the emergence of tool-use behaviours is probably larger than some researchers have suggested.

2. Does the Conclusion identify unresolved questions / gaps / future directions?
Yes, e.g., research into the role of genetics for the emergence of tool-use behaviours and the role of individual and social learning in other domains, such as social behaviours and gestures.

Additional comments

This is a very well-written manuscript which provides an overview which will be very useful for many researchers of several disciplines. I only have 2 points, which will be stated first, and the rest is just minor points. I am looking forward to seeing this work published soon and being discussed in the wider research community.

Table 1: Looking at the authors’ interpretation column, I get the impression that not that many papers make a strong claim for copying social learning to play a role in the findings of their respective studies. Many papers just seem to give a neutral description or actually acknowledge the possibility of individual learning. It would be quite insightful to try to classify the interpretations of the authors to see the proportion of interpretations in favour of the social learning hypothesis, those that stay neutral, and those that are compatible with the ZLS account. This can help identify whether the dominance of the social learning hypothesis is actually due to many researchers supporting it or whether it is more due to a few very influential researchers representing the social learning hypothesis. If the latter is the case, and it turns out that the social learning hypothesis is not that prevalent in the explanations given in the cited papers of the current review, this might actually support the case of the ZLS hypothesis and the arguments presented in the current review. This is not to say that the authors are wrong are in stating that there is a bias in the field towards the social learning hypothesis, this is probably the case. Yet, if the authors could show that the studies they found do not necessarily rely on the social learning hypothesis to interpret their findings, this could help shift readers’ perception of the field, which might encourage them to engage with the claims of the ZLS hypothesis and to more critically question the assumptions of the social learning hypothesis.

Discussion: The manuscript discusses the potential role of the environment, genetics, and the knowledge of pre-existing techniques. However, one crucial factor seems to be missing, which is other cognitive factors. This is especially important as the abstract and introduction introduce the focus on the mechanisms behind tool use behaviours. Of course, the focus of the current review is on individual and social learning, but they are only one part of the potential (cognitive) mechanisms involved. For example, previous literature has discussed the role of information processing and recombination, the motivation for object manipulation, physical cognition and causal reasoning, sensorimotor abilities, and executive functions (especially working memory), see references below. The review should, if not briefly discuss, at least point towards these other potentially crucial mechanisms underlying the acquisition and maintenance of tool-use behaviours.
References:
Call, J. (2013). Three ingredients for becoming a creative tool user. Tool use in animals. Cognition and Ecology, Crickette M. Sanz, Josep Call, & Christoph Boesch (eds.), Cambridge University Press.
Seed, A. & Byrne, R. (2010). Animal tool-use. Current Biology, 20, R1032-R1039.
Read, D. W. (2017). Quantitative Differences Between the Working Memory of Chimpanzees and Humans Give Rise to Qualitative Differences: Subitizing and Cranial Development.
Read, D. W. (2008).Working Memory: A Cognitive Limit to Non-Human Primate Recursive Thinking Prior to Hominid Evolution. Evolutionary Psychology. 6(4), 676-714. UCLA: Human Complex Systems. Retrieved from https://escholarship.org/uc/item/5d06v437

Minor points:

Lines 83-85: “A large number of non-human animal species (henceforth: animals) use tools (e.g., Finn, Tregenza, & Norman, 2009; Kenward et al., 2011; Krützen et al., 2005; Ottoni & Mannu, 2001; Robbins et al., 2016; Whiten et al., 1999).” The manuscript should here also refer to the comprehensive and fairly recent review of animal tool use by Shumaker et al. (2011), also because this review contains the currently widely accepted definition of tool use, which is not given in the current manuscript.
Reference:
Shumaker, R. W., Walkup, K. R., & Beck, B. B. (2011). Animal Tool Behavior. The Use and Manufacture of Tools by Animals. Revised and updated edition. The Johns Hopkins University Press.

Lines 83-85: “A large number of non-human animal species (henceforth: animals) use tools (e.g., Finn, Tregenza, & Norman, 2009; Kenward et al., 2011; Krützen et al., 2005; Ottoni & Mannu, 2001; Robbins et al., 2016; Whiten et al., 1999). Yet it is still unclear how these animals acquire and sustain their tool-use repertoires.” It seems like there is a sentence missing here linking the first and the second sentence: The first sentence makes a broad statement about tool use in any kind of animal species; the second sentence, by using the term “tool-use repertoires”, suggests that the narrower focus is on flexible tool behaviour and on those animals that have a repertoire of tool behaviours and not just a single, stereotyped tool use behaviour. Therefore, the use of “these animals” in the second sentence is not entirely correct – the question how the tool-use repertoires are acquired and sustained is probably not intended to apply to all tool-using animals, but to those capable of flexible tool behaviour. As Hunt et al. (2013) describe, there seem to be two classes of tool-use behaviour: 1) Stereotyped tool behaviour, which often occurs in a highly specific context with only little variation between individuals (often seen in fish and invertebrates, but see the octopus for a possible exception) and 2) flexible tool-use behaviour, which requires individual learning, is often not shown by all members of the population, and which is cognitively more demanding (tool use in birds and mammals). A brief addition to the text should be included explaining that the interesting explanandum are the mechanisms behind this latter class of tool-use behaviours.
Reference:
Hunt, G. R., Gray, R. D., & Taylor, A. H. (2013). Why is tool use rare in animals? In Crickette M. Sanz, Josep Call, & Christophe Boesch (eds.), Tool Use in Animals. Cognition and Ecology, 89-118.

Line 86: Extra space before “Early”, should be deleted.

Line 114: the term “behavioural forms” is a rather novel, technical term and should be defined here. Why behavioural forms instead of just behaviour?

Lines 113-115: “The ZLS proposes that many animal (and sometimes human) tool-use behavioural forms are within the ‘Zone of Latent Solutions’ of the species, and can be acquired through individual learning.” The manuscript provides no definition for the “Zone of Latent Solutions”. An easy solution could be to change the sentence to: “…, i.e., can be acquired through individual learning.”

Table 1: Searching the table would be a bit easier for the reader if within each species references were sorted alphabetically. I would suggest doing so, unless there was another factor the sources were sorted by (this should then be stated).

Table 1, first page, source Marshall-Pescini & Whiten (2008): 1. The comma after Marshall-Pescini should be replaced by a “&”. 2., and more importantly, it is unclear why this case was included. Even if it remains to be confirmed that Mawa had been living in deprived conditions, the authors suggested that he had already been familiar with nut cracking. So it is unclear on what basis this case is still included? This should be explained.
Similarly, why is Biro et al. (2013) included if there is the possibility that the chimpanzee had prior experience?

Table 1: Headings, “Author’s interpretation” – maybe change to “Authors’ interpretation”, as in most cases the papers are authored by more than a single person?

Throughout the document: maybe consider replacing the term “man-made” by a gender-neutral term, e.g., artificial, non-natural, human-made.

Line 289: “Wolfgang Köhler (1914-1920) spent many years investigating …” The dates in parentheses are misplaced, as their current position behind the name suggest them to be related to the lifetime. However, the dates are meant to specify the “many years” and so should be placed after years.

Line 294: “use sticks to retrieve honey from inside bee’s nests”. Apostrophe needs to be changed to “use sticks to retrieve honey from inside bees’ nests”.

Footnote 2: “could be taken to imply a sole, or primary, role of genetics in the emergence of a behaviour However, it is more” Full stop missing after “behaviour”.

Line 313, line 391, line 433: “Zimmerman, (2015),” “However, Zuberbühler et al., (1996)”, “ Kenward et al., (2005, 121)” – in all, remove comma before parentheses.

Line 464: “the authors argue that their “observations provethat innovative tool-related problem-solving” – add space to “provethat”

Lines 467-470: “Similarly, when naïve pigeons solved a task inspired by Köhler’s (1925) work with chimpanzees, in which the pigeons had to use boxes to reach a banana hanging outside their enclosure, the authors conclude that that they “have on hand an instance of insightful problem solving” (Epstein, 1985, 62).” Two issues: 1) Change “the authors” to “the author”; 2) remove one “that” from “conclude that that they”.

Line 545: “unlikely that the captivity effect enhances animal’s cognition” – apostrophe misplaced, should be “animals’ cognition”.

Line 583: “and, dolphin sponging behaviour” – remove comma after “and”.

Line 632: “Gruber et al.s’ work (2009)” change to “Gruber et al.’s (2009) work”

Line 653: “can be individual learnt” – change to “can be learnt individually”?

Line 668: “alongside recognising the equifinality of behaviours” – the role of the equifinality of behaviours for the reinnovation of a behaviour needs to be explained briefly.

Reviewer 2 ·

Basic reporting

This review of the role of individual learning in animal tool use will be of broad interest to those in animal cognition, comparative psychology and beyond. The review provides a comprehensive resource documenting tool uses in many animals, with the authors focusing on the role that individual, rather cultural processes may play in animals discoveries of tool uses. Overall, this will be a great addition to the literature, and it is clear that the authors put a lot of work into the article, reviewing over 200 papers!

I would like to see a few changes that should help introduce the subject matter to readers unfamiliar with animal tool use and the social learning literature:

1) Please define animal 'tool use'. There are many definitions in the literature and it would be helpful to include a short sentence detailing what criteria the authors are using. For example, does social and arbitrary tool use count, does tool use need to be goal directed etc?

2) Please define the social learning mechanisms included in the MS. For example, many readers will not know what stimulus or local enhancement mean. Perhaps include a table of definitions if necessary. Related to this, personally I find 'non-copying social learning' and 'copying social learning' a little confusing, especially as social learning mechanisms have been clearly defined in numerous reviews (e.g, Hoppitt and Laland). Perhaps it is best to refrain from introducing new terminology to an already quite complex area. Otherwise, I would like the authors to delineate which social learning mechanisms they see as falling under their two umbrella terms (e.g., is copying social learning just another term for 'imitation', and 'non-copying social learning' a term for all other social learning processes, and if so could this be renamed 'observational learning' or something else).

Minor points for the introduction:

L83. Perhaps cite Bentley-Condit & Smith 2010 and Beck 1980 as they also provide extensive reviews or catalogs of animal tool use.

L88. Could the authors note where the figure of 20 came from? General social information use is almost, if not ubiquitous in the animal kingdom, used even by solitary living animals (see papers by Dall/Giraldeau for examples of widespread social information use, and the behavioral ecology literature more broadly). Or is the 20 just relating to social learning of just 'tool use behaviors' - please clarify.

L97, Is 'copying' referring to 'imitation' here?

L87-97. The background given for academics interest in social learning, and in particular the role imitation plays in the emergence of animal behaviors, may need teasing out a little further. The authors suggest that research first looked at individual learning, then because some 'non-copying' social learning was documented, researchers hypothesized non-human animals may imitate like humans. I believe the background is a little more nuanced, where the first intimations of animal cultures received backlash since there were few direct examples of social learning in animals, despite regional variation in wild populations. More specifically, authors suggested that even if animals had cultures, they were incapable of the 'complex mechanisms' responsible for human cultures (e.g., Galef, 1992; see Kendal, 2008; Laland & Galef, 2009). This then drove the push to investigate whether animals transmitted information socially (it was already clear that individual learning occurs), whether traditions emerge in this way, as well as if other animals use mechanisms homologous to humans (the role of imitation).

L102-105. Please provide the references to back up the statement quoted below. Many studies actually find the opposite, documenting the propagation of novel behaviors within groups of primates and other animals via social processes. Unless the authors are using 'copy' to mean 'imitate behavior patterns' in which case the evidence is indeed more limited ( his again highlights the need for clear definitions of the terminology used in this paper).
"Experimental approaches that explicitly examined the ability to copy behaviours in unenculturated primates (i.e., animals who have not been exposed to extensive human contact and/or training; see also Henrich & Tennie, 2017) were unable to find spontaneous copying abilities in their subjects"

L100-L107. I'm not sure many researchers, if any, would suggest that learning tool use behaviors strictly requires imitation (or indeed that imitation is required at all). Rather, accounts of tool use often highlight the need for certain socioecological conditions e.g., the right ecology, motor control, mental capacity and social conditions (see van Schaik, Deaner & Merrill, 1999). Thus I'm not sure emphasis on individual and social learning is an 'alternative' view.

L104-105. Please explicitly define what is meant by enculturated and unenculturated here (or in the method section). This is particularly important as the methodology excludes enculturated animals from their survey. Do animals have to be hand reared to be enculturated, do they have to live in human environments, are zoo animals enculturated as they have extensive human contact and training? What about reserach facilities were animals can be tested many many times? Also why are the chimps at Yerkes considered enculturated (the Pope imitation study) and those at similar facilities not? I'm also unclear why the Paquette (1992) study (and indeed Kanzi) are discussed along with findings from unenculturated chimps (L292-297) when the Paquette paper notes their “chimpanzees were separated from their respective mothers very soon after birth, and were never again in the presence of adults of their species, they may be considered as good subjects for observation because they never experienced restrictive captivity conditions such as social isolation (see Menzel, Davenport & Rogers 1970). On the contrary, they evolved together in an environment rich in stimulations, benefiting from the care and attention of human surrogate mothers fulfilling their need for security.” Adding more detail concerning the criteria used would be very helpful.

L139. Perhaps remove "suggesting that animal individual learning abilities may have been underestimated in the current literature" given that the survey finds over 100 papers in which the authors state individual learning occurs. This seems quite a few.

Experimental design

No comment

Validity of the findings

The method is appropriate and the authors offer a well developed argument that meets the goals set out in the introduction.

Minor points:

Some of the statements made seem a little presumptuous and not necessarily supported by the literature they cite. For example, L243 notes "the authors interpret this 'instantaneous reinnovation' as evidence that the chimpanzee must have copied the behaviour by watching others beforehand". It is tricky to assert that this is a case of 'instantaneous reinvention' as the chimpanzees witnessed others perform the act before they themselves did so. Thus, it could be reinvention, socially learned, or a mix of both. It would probably be better to just state "the authors interpret this as evidence that.....". There are similar examples throughout the MS.

L244-246. Please note that if there is a lack of evidence with small samples and few studies testing an ability, it does not need to indicate that the ability is not present (lack of evidence does not equate to evidence of lack). Also, maybe address the conflicting evidence (e.g., Horner & Whiten, 2005) that showed copying causally irrelevant actions only when an opaque box was presented suggesting action ‘copying’ could be at play in wild born sanctuary chimps. (there are probably more examples).

L294-296. But don't wild chimps show hierarchical structuring of tool sequences (see Sanz and Morgan & Boesch papers), using 3-5 tool elements sets for extraction (tools serve different functions) and regional variation in this? I do not think that this complexity and tool behavior has been reproduced in naïve chimps, as far as I am aware. I think it would be nice if the authors balanced their argument by acknowledging this.

L300-307. Would focusing on some of the other examples tell a different story, for example, the authors recent study found that naive chimps could not reinvent nut cracking tool use behaviors that are evident in wild populations?

L315-316. Please provide a reference for ecological explanations for the variation seen in chimps ant-dipping. I believe Humle and Matsuzawa, 2002 is being referred to but since then more sophisticated analyses suggested there are both individual and socially learned elements (see Humle 2006; Mobuis et al., 2008; Schning et al., 2008).

L354-356. Were the capuchins in Westergaard & Suomi (1994) unenculturated? It may be worth making it clear to the reader that Kanzi is enculturated bonobo.

L423-425. I thought Hunt & Gray (2003) originally proposed cultural transmission may be important, and even went as far to suggest the NCC behavior as a potential case of cumulative cultural evolution.

L503. In saying "this review does not therefore suggest that non-coping social learning plays no role..." the authors are ruling out 'copying'. Does the review discount this as a possible mechanism? Perhaps change to "suggest that social learning plays no role..."

L521-523. There are also studies that do show an effect of model on the acquisition of tool behaviors (e.g., Whiten et 2005; Hopper studies with the Panpipes, among others), and their occurring faster than in groups that were not shown the tool use method. Perhaps direct the reader to some of these so they are aware of some alternative findings.

L547. As previously mentioned, please outline why these chimps (Pope et al., 2017) are enculturated and others with similar backgrounds are not.

L670-671. As the authors note, there is a role for social learning in tool use behaviors in addition to individual learning, and many of the studies in their Table 1 show evidence for both (social and asocial learning). So it seems a little strong saying 'this review suggests instead that it is perhaps time to include individual learning as a serious alternative" as this suggests discounting evidence for social learning as individual learning also occurs. Perhaps the review suggests that the role of individual learning needs to be taken more seriously, or that researchers should acknowledge it is often a mix of the two processes that are important.

---

## Round 0.2 · Minor Revisions

Thank you for your revisions. I think the goal of the paper and the significance of that aim are much clearer now. I think the paper will be a very useful resource for researchers interested in social learning and in tool use. I do have a few minor suggestions to further clarify the writing in a few places but I expect these to be relatively minor changes that will not take a great degree of time or effort.
All line numbers refer to the reviewing pdf.
I think the definition of enculturated could use some work. There is quite a bit of controversy over how best to classify especially apes as such. Although some might agree with the definition provided on lines 121-123, many would argue for a more nuanced definition that refers to the extent to which the apes are emerged in human culture and raised in ways similar to how human parents would raise their own children. Most lab and zoo apes experience significant human training and contact but would not be considered enculturated. Think Sue Savage Rumbaugh’s bonobos versus Tomasello or Povinelli’s chimpanzees.
I think it would be better to provide definitions for key terms in the text itself. You can also refer to the table but I don’t think the reader should have to continually refer to the table while reading the introduction in order to understand, for example, distinctions between copying and non-copying social learning, which are certainly not obvious.
I don’t like the definition of emulation as copying social learning given that emulation is by definition, the actor NOT copying the exact behavioral form to arrive at a similar outcome.
I notice that you do not distinguish between “true imitation” and “copying.” Imitation typically requires the understanding of the goal or the intent of the actor to be copied, does it not? How important is that here? I think these definitions are much too fraught with confusion in the literature to be casually glossed over here. Typically, it is also important that the behavior is novel, displayed in a novel context, or within a novel sequence to qualify as having been learned through imitation. Given that the behavior itself is unlikely to be completely outside of the organism’s behavioral repertoire, these qualifications are important.
I am intrigued by your suggestion that researchers are more likely to attribute tool use in primates to social learning, compared to the individual learning hypotheses advocated for in birds. Could you speculate as to why that might be the case?
You might consider the following paper on tool use in gorillas as well:
Luef, E. M., & Heschl, A. (2017). Triadic interactions with tools in a gorilla. Animal Behavior and Cognition, 4(2), 136–145. https://doi.org/10.12966/abc.01.05.2017
Please double-check the interpretation of Epstein (1985) cited on line 510. It is my impression that Epstein’s work typically shows how apparently “higher order” cognitive processes, such as insight, can better be explained using associative mechanisms.
Minor Points:
Line 99 throughout is misspelled.
Capitalize Table when referring to a table throughout.
The term “artefact form” does not appear in Table 1 (see line 103). Can you clarify use of this term here please?
Line 122, I think you mean e.g. not i.e.
The ordering of alternatives on lines 113-114 is a bit odd. You should move from most exclusive to least inclusive or vice versa.
Watch the placement of “only.” ON line 479, it should be “exist only in captivity..”
The “Although” at the beginning of the sentence beginning on line 713 doesn’t seem appropriate.
Check formatting of the references carefully. There are some cases of improper capitalization in journal titles.

---

## Round 0.3 · accepted · Accept

Thank you very much for attending to these last minor revisions. And thank you for a nice contribution to the literature on social learning.